# HoP: Homeomorphic Polar Learning for Hard Constrained Optimization

## Abstract

Constrained optimization demands highly efficient solvers, which promotes the development of learn-to-optimize (L2O) approaches. As a data-driven method, L2O leverages neural networks to efficiently produce approximate solutions. However, a significant challenge remains in ensuring both optimality and feasibility of neural network's output. To tackle this issue, we introduce Homeomorphic Polar Learning (HoP) to solve the hard-constrained optimization by embedding a homeomorphic mapping in neural networks. The bijective structure enables end-to-end training without extra penalty or correction. For performance evaluation, we evaluate HoP's performance across a variety of synthetic optimization tasks and real-world applications in wireless communications. Across synthetic and real tasks, HoP achieves zero violations while remaining competitive in optimality and significantly faster than classical solvers: on polygon-constrained sinusoidal QP it matches SLSQP within $16\times$ speedup; on high-dimensional semi-unbounded problems it is tens of times faster than the optimizer with comparable or better objectives; and on QoS-MISO WSR it preserves $0\%$ violations with $11\times$ speedup over SCS+FP. These results demonstrate that HoP provides a practical, general, and strictly feasible alternative to penalty-based or projection-based L2O methods.

## 1 Introduction

Constrained optimization plays a fundamental role in numerous scientific and engineering applications (Liu et al., 2024; Abido, 2002; Rockafellar & Uryasev, 2013), while it suffers from significant computation overhead with traditional solvers. To address these limitations, the paradigm of learn-to-optimize (L2O) has emerged as a promising alternative (Hounie et al., 2024; Ding et al., 2024). As a learning-based scheme, L2O takes optimization parameters as the inputs of the neural network (NN), which can efficiently obtain approximate solutions from the outputs. Despite its advantages, L2O faces significant challenges as NN lacks guarantees to strictly output solutions within the feasible region defined by the constraints (Donti et al., 2021). Current works have attempted to address this limitation through various approaches, including supervised learning (Zamzam & Baker, 2020), incorporating constrained violations into the loss function (Zhang et al., 2024; Xu et al., 2018), post-correction methods (Donti et al., 2021), implicit differentiation (Amos & Kolter, 2017), and other techniques (Zhong et al., 2023; Misra et al., 2022). However, these methods often exhibit limitations in optimal solution searching and hard constraints violation control.

We propose the homeomorphic polar learning (HoP) to address hard constraints optimization. HoP is a learning-based framework inspired by principles of measure transformation and topological structure induced by constraints, designed to ensure both the feasibility and optimality of solutions. As illustrated in Fig. 1, the problem parameters are fed into a neural network, whose raw outputs are transformed by bounded functions into polar sphere vectors. Subsequently, through the proposed homeomorphic mapping, the polar sphere vectors are mapped to Cartesian coordinates, strictly following to the original constraints. Furthermore, HoP is trained end-to-end with the objective function as the direct training loss, thereby avoiding optimality gap introduced by composite loss. The key contributions are:

- **Novel formulation of hard-constrained optimization via polar coordinates and homeomorphic mapping**. HoP is the first L2O framework that uses polar coordinates to solve hard constrained

problems. It maps an arbitrary solution space into a deterministic n-ball, eliminating optimality bias introduced by post-correction or penalty based methods.

- **Reconnection strategies for polar optimization**. To address challenges specific to polar coordinate optimization, such as radial stagnation and angular freezing, we propose geometric reconnection strategies. We provide rigorous theoretical analyses to validate the stability of these solutions.

- **Superior experimental performance with zero violation.** Through extensive ablation and comparative experiments, we validate the feasibility and optimality of our approach across a wide range of problems. Results consistently show that HoP outperforms both traditional and learning-based solvers in terms of constraint satisfaction and optimization efficiency.

## 2 RELATED WORK

We present related works for constrained optimization problems using L2O. Broadly, researches in this area can be categorized into two directions: soft and hard constrained L2O.

### 2.1 SOFT CONSTRAINED OPTIMIZATION WITH L2O

Soft constrained L2O emphasizes on enhancing computational efficiency on NN inference speed while tolerating a limited rate of constraint violations. Early research in this domain explored the use of supervised learning (SL) to directly solve optimization problems, where the optimal variable $\mathbf{y}^*$ is provided as labels by optimizer

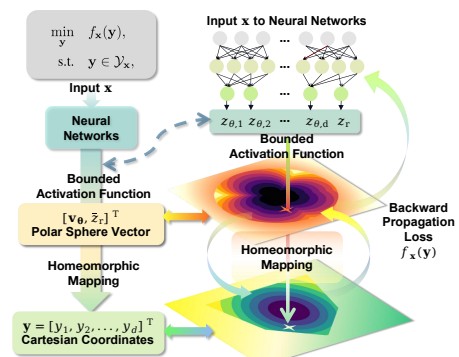

Figure 1: The flow structure of HoP framework

(Zamzam & Baker, 2020; Guha et al., 2019). Another prominent direction involves incorporating constraint violations into the objective function by Karush-Kuhn-Tucker conditions (Donti et al., 2021; Zhang et al., 2024; Xu et al., 2018). In this methods, constraints are reformulated as penalty terms and integrated into the objective function as the loss for self-supervised learning (SSL). Subsequent advancements introduced alternative optimization based learning, where variables and multipliers are alternately optimized through dual NNs (Park & Van Hentenryck, 2023; Kim & Kim, 2023; Nandwani et al., 2019). More recent related researches include preventive learning, which incorporates pre-processing in learning to avoid violation (Zhao et al., 2023). Additionally, resilience-based constraint relaxation methods dynamically adjust constraints throughout the learning process to balance feasibility and overall performance (Hounie et al., 2024; Ding et al., 2024). Nevertheless, real-world optimization problems often demand strict feasibility which causes significant deployment challenges for soft-constrained L2O techniques.

### 2.2 HARD CONSTRAINED OPTIMIZATION WITH L2O

Hard constrained optimization in L2O prioritizes strict adherence to constraints, even if it results in reduced optimality or slower computation speed. Traditional optimization methods often employ proximal optimization techniques to guarantee feasibility (Cristian et al., 2023; Min et al., 2024). Early methods also used activation functions to enforce basic hard constraints (Sun et al., 2018). Implicit differentiation became a popular approach for effectively handling equality constraints (Amos & Kolter, 2017; Donti et al., 2021; Huang et al., 2021). However, inequality constraints typically require additional correction steps, which compromise optimality to search for suboptimal feasible solutions (Donti et al., 2021). An alternative strategy proposed by Li et al. (2023) utilized the geometric properties of linear constraints to ensure outputs within the feasible region, although this method is limited to linear constraints. A related study by Liang et al. (2023) introduced a novel topological perspective for constrained learning, while the presence of distortion and nonstationary constraints undermine feasibility guarantees. Other studies such as Misra et al. (2022); Guha et al. (2019) focused on eliminating redundant constraints to improve inference speed instead of solving optimization problem by NNs directly. In certain physical applications, discrete mesh-based approaches restrict feasible solutions to predefined points on a mesh (Amos et al., 2017; Zhong et al., 2023; Négiar et al.,

2022). Although these methods strictly enforce feasibility, they often lack optimality or flexibility in general scenarios.

## 3 METHODOLOGY

At a high level, HoP replaces penalties and post-hoc projections with geometry. We compose a NN with an explicit, differentiable bijection from a normalized polar parameterization to the feasible set. Feasibility is therefore guaranteed by construction, and the training loss can be the true objective. According to the designing, neither heuristic tuning of penalty weights nor projection-induced bias is required, and gradients point where the objective dictates.

We develop the construction progressively. We begin with a 1-D warm-up that captures the essence of hard constrained feasibility via bounded reparameterization, lift the idea to 2-D convex sets where a radial map from an interior point yields a simple homeomorphism, and then generalize to high dimensions and semi-unbounded domains via a spherical-angle parameterization. Along the way, we analyze continuity, bijectivity, and Jacobian scaling near the boundary, and we introduce a reconnection mechanism that eliminates the stagnation endemic to polar parameterizations.

### 3.1 HOMEOMORPHIC POLAR LEARNING

**Problem formulation and design goals.** We consider the following optimization problem:

$$\min_{\mathbf{y}} \quad f_\mathbf{x}(\mathbf{y}), \quad \text{s.t.} \quad \mathbf{y} \in \mathcal{Y}_\mathbf{x}, \tag{1}$$

where $f_\mathbf{x} : \mathbb{R}^n \to \mathbb{R}$ is (possibly) nonconvex and differentiable, and $\mathcal{Y}_\mathbf{x}$ is defined as a convex constraint set, both parameterized by $\mathbf{x}$. Unless otherwise stated, we assume $\mathcal{Y}_\mathbf{x}$ is convex and admits at least one interior point; To facilitate that the outputs of a learning model satisfy the given constraints, we construct a mapping from the raw output space to a constrained space with homeomorphic transformation. To illustrate this approach, we begin by introducing the core concept behind HoP, the homeomorphic mappings, which leverage the mathematical properties of homeomorphisms and are formally defined as follows:

**Definition 3.1.** Let $X = (S_X, \mathcal{T}_X)$ and $Y = (S_Y, \mathcal{T}_Y)$ be two topological spaces, where: (1) $S_X$ and $S_Y$ are point sets; (2) $\mathcal{T}_X$ and $\mathcal{T}_Y$ are topologies on $S_X$ and $S_Y$, respectively. Then function $\mathcal{H} : X \to Y$ is called a homeomorphism if and only if $\mathcal{H}$ is a bijection and continuous with respect to the topologies $\mathcal{T}_X$ and $\mathcal{T}_Y$, while its inverse function $\mathcal{H}^{-1} : Y \to X$ exists and is also continuous.

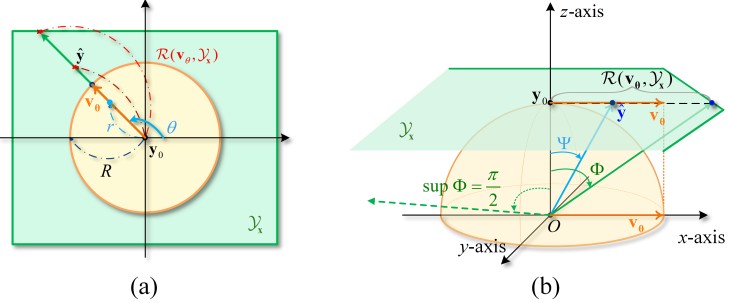

(a)             (b)

Figure 2: (a) *Illustration of the 2-D HoP principle.* The feasible region $\mathcal{Y}_\mathbf{x}$ is depicted in green and $\mathbf{y}_0 \in \mathcal{Y}_\mathbf{x}$. We construct a unit circle (yellow region) centered at $\mathbf{y}_0$ and parameterize it in polar coordinates. Typically, $\mathbf{y}_0$ is obtained by solving a convex optimization problem over $\mathcal{Y}_\mathbf{x}$, like computing the Chebyshev center, which maximizes the minimum distance from the point to the boundary of $\mathcal{Y}_\mathbf{x}$ (b) *Sketch of the spherical coordinate transformation for semi-unbounded constraints.* The "blue" ray at angle $\psi$ intersects the green plane at $\hat{\mathbf{y}}$, staying strictly within the feasible cone, while the limiting "green" ray at $\phi$ meets the boundary.

**The 1-D Case** The most reliable way to enforce the constraint $a < y < b$ is by parameterizing the interval rather than penalizing violations. Let $\mathcal{Y}_\mathbf{x} = (a, b)$. We map the network logit $z \in \mathbb{R}$ to a feasible $\hat{y}$ via a smooth, strictly monotone, bounded function $\mathcal{B}$ followed by an affine rescaling:

$$\mathcal{H} : \quad \hat{y} = a + \mathcal{B}(z)(b - a). \tag{2}$$

Because $\mathcal{B} : \mathbb{R} \to (0, 1)$ is bijective with a continuous inverse (e.g., the logit of a sigmoid), Eq. 2 is an homeomorphism between $\mathbb{R}$ and $(a, b)$ (Def. 3.1). Feasibility is guaranteed by construction, and gradients of the true objective $f_{\mathbf{x}}(\hat{y})$ backpropagate through $\mathcal{H}$ without the bias introduced by penalties or post-hoc projections. If the optimizer resides on the boundary, $\hat{y}$ can approach it arbitrarily closely from the interior.

**The 2-D case** We now lift the construction to a 2-D convex feasible set $\mathcal{Y}_{\mathbf{x}} \subset \mathbb{R}^2$. Choose an interior reference point $\mathbf{y}_0 \in \mathrm{int}(\mathcal{Y}_{\mathbf{x}})$ (e.g., the Chebyshev center; any interior point suffices). For a unit direction $\mathbf{v}_\theta$ on the circle, define the support radius

$$\mathcal{R}(\mathbf{v}_\theta, \mathcal{Y}_{\mathbf{x}}) \;=\; \sup\{\, t \geq 0 : \; \mathbf{y}_0 + t\,\mathbf{v}_\theta \in \mathcal{Y}_{\mathbf{x}} \,\},$$

which is finite and unique for convex $\mathcal{Y}_{\mathbf{x}}$ specified by $\mathbf{v}_\theta$. The 2-D HoP map sends a normalized polar pair $(r, \theta) \in (0, 1) \times (0, 2\pi)$ to a point in the interior (see Fig. 2a):

$$\mathcal{H} : \quad \hat{\mathbf{y}} \;=\; \mathbf{y}_0 \;+\; r\,\mathbf{v}_\theta\,\mathcal{R}(\mathbf{v}_\theta, \mathcal{Y}_{\mathbf{x}}). \tag{3}$$

Intuitively, $\theta$ chooses a ray and $r$ interpolates along that ray from $\mathbf{y}_0$ to the boundary; convexity ensures each ray intersects the boundary exactly once, which yields homeomorphism between $(0, 1) \times (0, 2\pi)$ and $\mathrm{int}(\mathcal{Y}_{\mathbf{x}})$. The inverse is explicit: for any $\hat{\mathbf{y}} \neq \mathbf{y}_0$, $\mathbf{v}_\theta = (\hat{\mathbf{y}} - \mathbf{y}_0)/\|\hat{\mathbf{y}} - \mathbf{y}_0\|_2$ and $r = \|\hat{\mathbf{y}} - \mathbf{y}_0\|_2/\mathcal{R}(\mathbf{v}_\theta, \mathcal{Y}_{\mathbf{x}})$.

To couple this geometry with a network, we produce $(\theta, r)$ from raw outputs $\mathbf{z} = [z_\theta, z_r]^\top$ using bounded activations and the standard polar unit vector:

$$\begin{bmatrix} \theta \\ r \end{bmatrix} = \begin{bmatrix} 2\pi & 0 \\ 0 & 1 \end{bmatrix} \begin{bmatrix} \mathcal{B}(z_\theta) \\ \mathcal{B}(z_r) \end{bmatrix}, \qquad \mathbf{v}_\theta = \begin{bmatrix} \cos\theta \\ \sin\theta \end{bmatrix}. \tag{4}$$

Composing the NN with $\mathcal{H}$ thus hard-encodes feasibility: every forward pass lies strictly inside $\mathcal{Y}_{\mathbf{x}}$, and the training signal is the true objective $f_{\mathbf{x}}(\hat{\mathbf{y}})$. Computationally, $\mathcal{R}(\mathbf{v}_\theta, \mathcal{Y}_{\mathbf{x}})$ is a 1-D search along a ray. This search is often tractable in closed form for simple constraints, and can otherwise be resolved efficiently via binary search in convex domains. Fig. 2(a) visualizes the geometry that underpins the mapping.

**Extension to the semi-unbounded and high-dimensional case** In semi-unbounded problems the feasible set $\mathcal{Y}_{\mathbf{x}}$ extends to infinity along some directions, so the support radius $\mathcal{R}(\mathbf{v}_\theta, \mathcal{Y}_{\mathbf{x}})$ used in the bounded 2-D case may be infinite. A direct generalization of spherical angles is also unattractive in high dimensions: classical $d$-dimensional spherical coordinates require $(d - 1)$ nested $\{\sin, \cos\}$ terms, introduce chart singularities at the poles, and accumulate unnecessary trigonometric cost as $d$ grows. We avoid both issues with an angle-free sphere parameterization for direction and a single scalar angle for range. Given NN outputs $\mathbf{z} = [\mathbf{z}_\theta, z_r]^\top$ with $\mathbf{z}_\theta \in \mathbb{R}^d$ and $z_r \in \mathbb{R}$, we map to a unit direction and a bounded range coefficient by

$$\bar{z}_r \;=\; \mathcal{B}(z_r), \tag{5}$$
$$\mathbf{v}_\theta \;=\; (\mathbf{z}_\theta + \varepsilon)/(\|\mathbf{z}_\theta\|_2 + \varepsilon), \tag{6}$$

---

**Algorithm 1: HoP: homeomorphic polar learning**

Prepare parameters $\mathcal{X}_{\mathrm{train}}$ and $\mathcal{X}_{\mathrm{test}}$, initialize NN, polar center $\mathbf{y}_0$ for every instance in $\mathcal{X}_{\mathrm{train}}$ and $\mathcal{X}_{\mathrm{test}}$.

TRAIN:
**for** every epoch **do**
    **for** every batch data $\mathbf{x}$ in $\mathcal{X}_{\mathrm{train}}$ **do**
        Update $[\mathbf{z}_\theta, z_r]^T = \mathrm{NN}(\mathbf{x})$;
        Update $\mathbf{v}_\theta, \bar{z}_r$ by Eqs. (13);
        Update $\psi$ using Eq. (7)
        Output estimated variables $\hat{\mathbf{y}}$ by Eq. (8);
        Update loss function $\mathcal{L}(\hat{\mathbf{y}})$;
        Backward propagation to update NN parameters.
    **end for**
**end for**
TEST:
**for** every batch data $\mathbf{x}$ in $\mathcal{X}_{\mathrm{test}}$ **do**
    Update $[\mathbf{z}_\theta, z_r]^T = \mathrm{NN}(\mathbf{x})$;
    Update $\mathbf{v}_\theta$ and $\bar{z}_r$ by Eqs. (13);
    Update $\psi$ using Eq. (7)
    Output estimated variables $\hat{\mathbf{y}}$ by Eq. (8);
    Compute the gap between $\mathcal{L}(\hat{\mathbf{y}})$ and $\mathcal{L}(\mathbf{y}^*)$;
**end for**

---

where $\varepsilon > 0$ is a small number which protects against an undefined normalization when $\mathbf{z}_\theta = 0$. We then parameterize semi-unbounded rays by a bounded angle rather than a possibly unbounded distance. For any direction $\mathbf{v}_\theta$, define the boundary angle

$$\phi \;=\; \tan^{-1}\big(\mathcal{R}(\mathbf{v}_\theta, \mathcal{Y}_{\mathbf{x}})\big) \in (0, \tfrac{\pi}{2}],$$

with the convention $\phi = \frac{\pi}{2}$ when the set is unbounded along $\mathbf{v}_\theta$. Let the learned angle be a fraction of this bound,

$$\psi = \bar{z}_r \phi \qquad \left(\text{optionally } \psi \leq \phi - \epsilon \text{ with a tiny margin } \epsilon > 0\right), \tag{7}$$

and map back to Cartesian coordinates by

$$\mathcal{H}: \quad \hat{\mathbf{y}} = \mathbf{y}_0 + \mathbf{v}_\theta \tan(\psi). \tag{8}$$

Since $\tan(\cdot)$ is strictly increasing on $(0, \frac{\pi}{2})$ and $0 \leq \psi < \phi \leq \frac{\pi}{2}$, along each ray we have

$$0 \leq \|\hat{\mathbf{y}} - \mathbf{y}_0\|_2 = \tan(\psi) < \tan(\phi) = \mathcal{R}(\mathbf{v}_\theta, \mathcal{Y}_\mathbf{x}),$$

so $\hat{\mathbf{y}}$ lies strictly inside $\mathcal{Y}_\mathbf{x}$ on every feasible ray. When the ray is unbounded, $\phi = \frac{\pi}{2}$ and $\tan(\psi)$ remains finite, again placing $\hat{\mathbf{y}}$ in the interior. The direction $\mathbf{v}_\theta$ is obtained by a single normalization (no $(d-1)$ trigonometric angles, no pole singularities), and the range uses only the scalar $\tan(\psi)$. Thus, HoP's semi-unbounded map is dimension-agnostic in its trigonometric cost and numerically well-behaved. The mapping is homeomorphism with an explicit inverse $\mathbf{v}_\theta = (\hat{\mathbf{y}} - \mathbf{y}_0)/\|\hat{\mathbf{y}} - \mathbf{y}_0\|_2$, $\psi = \tan^{-1}\|\hat{\mathbf{y}} - \mathbf{y}_0\|_2$. See Fig. 2(b) for a 2-D example.

**Jacobian control** The Jacobian determinant of $\hat{\mathbf{y}}(\psi, \mathbf{v}_\theta)$ is $\det D\hat{\mathbf{y}} = \tan^{d-1}\psi \; \sec^2\psi$, which diverges only as $\psi \to \frac{\pi}{2}$. Enforcing a tiny angular margin $\psi \leq \phi - \epsilon$ bounds this distortion and yields stable gradients; see Appendix A for the full derivation and bounds.

## 3.2 RESOLVING STAGNATION IN POLAR OPTIMIZATION VIA RECONNECTION

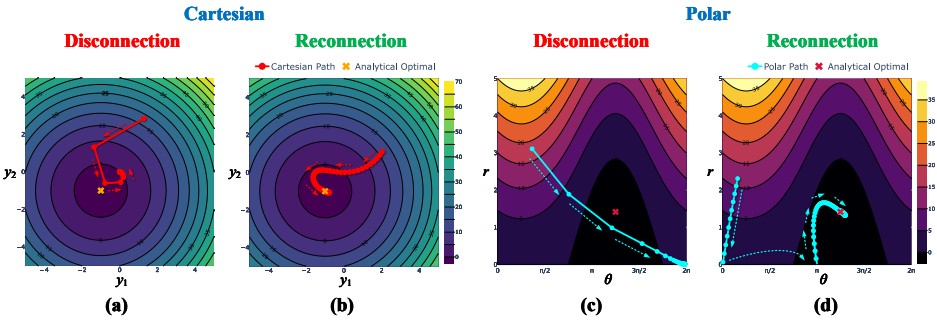

Figure 3: Comparison of optimization trajectories in Cartesian and polar coordinates. The (a) and (c) demonstrate disconnection issues in polar coordinates, while (b) and (d) show improved behavior with reconnection strategies.

In order to reveal the stagnation phenomenon in polar optimization, we consider a simple minimization problem $f : \mathbb{R}^n \to \mathbb{R}$ with variable $\mathbf{y} \in \mathbb{R}^n$ where the $\mathbf{y}_0$ is origin. Since reconnection design is constraint-agnostic, the problem is simplified as unbounded problem. We apply a polar decomposition $\hat{\mathbf{y}} = r\,\mathbf{v}_\theta$, $r \in \mathbb{R}_{\geq 0}$ and $\mathbf{v}_\theta \in \mathbb{S}^{n-1} := \{\mathbf{u} \in \mathbb{R}^n : \|\mathbf{u}\|_2 = 1\}$. This induces a product-space geometry $(0, \infty) \times \mathbb{S}^{n-1} \to \mathbb{R}^n \setminus \{\mathbf{0}\}$. Then the gradient of $f(r\,\mathbf{v}_\theta)$ is decomposed as:

$$\frac{\partial f}{\partial r} = \langle \nabla f(\mathbf{y}), \mathbf{v}_\theta \rangle, \qquad \text{grad}_{\mathbf{v}_\theta} f = r\,\Pi_{\mathbf{v}_\theta}^\perp \nabla f(\mathbf{y}), \tag{9}$$

where $\Pi_{\mathbf{v}_\theta}^\perp := I - \mathbf{v}_\theta \mathbf{v}_\theta^\top$ projects onto $T_{\mathbf{v}_\theta}\mathbb{S}^{n-1}$. The spherical component vanishes as $r$ approaching to 0, regardless of gradient magnitude. Standard gradient descent on $\mathbb{R}_{\geq 0} \times \mathbb{S}^{n-1}$ with step size $\eta_t$ proceeds as:

$$\tilde{r}_{t+1} = r_t - \eta_t \langle \nabla f(\mathbf{y}_t), \mathbf{v}_{\theta,t} \rangle, \qquad r_{t+1} = \max\{0, \tilde{r}_{t+1}\}, \tag{10}$$

$$\tilde{\mathbf{v}}_{\theta,t+1} = \mathbf{v}_{\theta,t} - \eta_t r_{t+1} \Pi_{\mathbf{v}_{\theta,t}}^\perp \nabla f(\mathbf{y}_t), \qquad \mathbf{v}_{\theta,t+1} = \frac{\tilde{\mathbf{v}}_{\theta,t+1}}{\left\|\tilde{\mathbf{v}}_{\theta,t+1}\right\|_2}. \tag{11}$$

The gradient descent update rule creates a fundamental stagnation phenomenon in the polar coordinates. Fig. 3(a) and (c) illustrates this through optimization trajectories on a quadratic objective with minimum away from the origin. When the optimization trajectory, viewed in Cartesian coordinates, would pass near or through the origin on its way to the optimum, polar-parameterized gradient descent exhibits a critical failure at the origin due to its geometric constraints.

The stagnation arises from the interaction between the gradient structure and the non-negativity constraint $r \geq 0$. When the radial gradient $\langle \nabla f(\mathbf{y}_t), \mathbf{v}_{\theta,t} \rangle > 0$ points inward (toward decreasing $r$), the optimizer wants to pass through the origin to continue along the opposite ray. However, the projection $r_{t+1} = \max\{0, \tilde{r}_{t+1}\}$ prevents this passage. As the iteration approach $r = 0$, the spherical gradient component $r\Pi_{\mathbf{v}_{\theta,t}}^{\perp} \nabla f(\mathbf{y}_t)$ vanishes, eliminating any ability to adjust direction. Once $r_{t+1} = 0$, we have $\tilde{\mathbf{v}}_{\theta,t+1} = \mathbf{v}_{\theta,t} - 0 = \mathbf{v}_{\theta,t}$, permanently freezing the direction. The algorithm enters an absorbing state where it repeatedly attempts to cross the origin but is projected back to $(\mathbf{y}, \mathbf{v}_\theta) = (\mathbf{0}, \mathbf{v}^*)$, even though $\nabla f(\mathbf{0}) \neq \mathbf{0}$ may point in a completely different direction $\theta^*$. We term this geometric artifact **stagnation** as a fixed point induced by the parameterization rather than the objective function itself.

To eliminate this stagnation, we propose **geometric reconnection**. Instead of projecting to zero, we allow signed radii and apply reflection when crossing the origin. For $\tilde{r}_{t+1} < 0$, we set

$$(r_{t+1}, \mathbf{v}_{\theta,t+1}) = (|\tilde{r}_{t+1}|, -\tilde{\mathbf{v}}_{\theta,t+1}/\|\tilde{\mathbf{v}}_{\theta,t+1}\|_2), \tag{12}$$

in a flip direction while preserving continuity since $r\mathbf{v} = |r|(-\mathbf{v})$. The direction update then uses the reconnected direction $\mathbf{v}_{\theta,t}^{\mathrm{rec}}$. After crossing with the flip, the spherical gradient remains nonzero, enabling continued adaptation. As shown in Fig. 3(b), reconnection allows smooth traversal through the origin, matching Cartesian performance (Fig. 3(d)) while maintaining the benefits of polar parameterization. As a consequence, to implement a similar reconnection design in Eq. (12), we reformulate the Eqs. (5) and (6) as follows

$$\bar{z}_r = \mathcal{B}(|z_r|), \quad \mathbf{v}_\theta = \begin{cases} (\mathbf{z}_\theta + \varepsilon)/(\|\mathbf{z}_\theta\|_2 + \varepsilon), & \text{if } z_r \geq 0, \\ -(\mathbf{z}_\theta + \varepsilon)/(\|\mathbf{z}_\theta\|_2 + \varepsilon), & \text{otherwise.} \end{cases}, \tag{13}$$

Finally, the complete HoP procedure is summarized in Algorithm 1.

## 4 EXPERIMENTS

In following experiments we evaluate HoP by comparing with other methods in three criteria, including optimality, feasibility and computation efficiency. Comparative experiments with traditional optimizers, DC3 (Donti et al., 2021), HomeoProj (Liang et al., 2023) and other NN-based L2O approaches are conducted to validate HoP's effectiveness. All NNs follow a uniform architecture: a 3-layer multilayer perceptron (MLP) with ReLU activation functions. Since other NN-based methods such as DC3 typically rely on penalty functions to handle constraints, where the penalty is defined as $\mathcal{P}(\hat{\mathbf{y}}, \mathcal{Y}_\mathbf{x}) = \mathbb{I}(\hat{\mathbf{y}} \notin \mathcal{Y}_\mathbf{x}) \cdot \mathrm{dist}(\hat{\mathbf{y}}, \mathcal{Y}_\mathbf{x})$, where $\mathbb{I}(\hat{\mathbf{y}} \notin \mathcal{Y}_\mathbf{x})$ indicates constraint violations, and $\mathrm{dist}(\hat{\mathbf{y}}, \mathcal{Y}_\mathbf{x})$ quantifies the distance to the constraint set.

- Optimizer: For synthetic benchmarks (Experiments 4.1), the Sequential Least Squares Programming (SLSQP) is used as the solver. For the quality-of-service-aware multi-input-single-output communication system weighted sum rate (QoS-MISO WSR) problem in Experiment 4.2, Splitting Conic Solver (SCS) (Diamond & Boyd, 2016) and fractional programming (FP) (Shen & Yu, 2018) is applied for alternative optimization as the solver baseline.

- HoP (+R): The proposed method with the reconnection.

- HoP (−R): The ablation baseline of the proposed method excluding the reconnection.

- NN-SSL (Self-Supervised Learning): The loss function consists individually of the objective function without any penalty term, defined as $\mathcal{L}(\hat{\mathbf{y}}) = f_\mathbf{x}(\hat{\mathbf{y}})$.

- NN-SL: The loss minimizes the mean squared error (MSE) between predictions $\hat{\mathbf{y}}$ and the target labels $\mathbf{y}^*$, expressed as $\mathcal{L}(\hat{\mathbf{y}}) = (\hat{\mathbf{y}} - \mathbf{y}^*)^2$.

- NN-SL-SC (Soft-Constraint + Supervised Learning): The loss function integrates the MSE and a soft constraint penalty, expressed as $\mathcal{L}(\hat{\mathbf{y}}) = (\hat{\mathbf{y}} - \mathbf{y}^*)^2 + \lambda\mathcal{P}(\hat{\mathbf{y}}), \lambda \geq 0$.

- NN-SSL-SC (Soft-Constraint + Self-Supervised Learning): The loss incorporates both the objective function and a soft constraint penalty: $\mathcal{L}(\hat{\mathbf{y}}) = f_\mathbf{x}(\hat{\mathbf{y}}) + \lambda\mathcal{P}(\hat{\mathbf{y}})$.

- DC3: This baseline follows Donti et al. (2021), using the soft-constraint loss of NN-SSL-SC with additional gradient post-corrections to guarantee feasibility.

Table 1: Experimental results over synthetic benchmarks and real engineering problem

| Problem | Methods | Obj. Value ↓ | Max. Cons ↓ | Mean. Cons ↓ | Vio. Rate ↓ | Time / ms ↓ |
|---|---|---|---|---|---|---|
| **Polygon constrained problem** | Optimizer | −29.7252 | 0.0000 | 0.0000 | 0.00% | 0.07104 |
| | HoP (+R) | **−29.7170** | 0.0000 | 0.0000 | 0.00% | 0.00398+0.00046 |
| | HoP (−R) | −0.8133 | 0.0000 | 0.0000 | 0.00% | **0.00398+0.00045** |
| | NN-SSL | −29.7481 | 0.0334 | 0.0029 | 18.10% | 0.00004 |
| | NN-SL | −29.7247 | 0.0153 | 0.0001 | 7.47% | 0.00004 |
| | NN-SSL-SC | −29.5601 | 0.0108 | 0.0001 | 2.94% | 0.00004 |
| | NN-SL-SC | −29.7203 | 0.0080 | 0.0001 | 0.35% | 0.00004 |
| | DC3 | −29.6893 | 0.0000 | 0.0000 | 0.00% | 0.01491 |
| | HomeoProj | −29.3131 | 0.0000 | 0.0000 | 0.00% | 0.02778 |
| **Non-convex $\ell_p$-norm problem** | Optimizer | −0.4824 | 0.0000 | 0.0000 | 0.00% | 1.49371 |
| | HoP (+R) | **−0.3886** | 0.0000 | 0.0000 | 0.00% | offline+0.00946 |
| | HoP (−R) | −0.3001 | 0.0000 | 0.0000 | 0.00% | offline+**0.00887** |
| | NN-SSL | −1.2227 | 67.6524 | 17.5511 | 100.00% | 0.00886 |
| | NN-SL | −0.5285 | 10.2304 | 2.1575 | 99.96% | 0.00886 |
| | NN-SSL-SC | −0.0679 | 0.4356 | 0.0007 | 0.75% | 0.00886 |
| | NN-SL-SC | −0.0132 | 0.3018 | 0.0002 | 0.02% | 0.00886 |
| | DC3 | −0.0730 | 0.0000 | 0.0000 | 0.00% | 0.02295 |
| | HomeoProj | −0.0345 | 0.0000 | 0.0000 | 0.00% | 0.18021 |
| **High-dimensional semi-unbounded problem** | Optimizer | −197.3113 | 0.0000 | 0.0000 | 0.00% | 4.87273 |
| | HoP (+R) | **−198.3798** | 0.0000 | 0.0000 | 0.00% | 0.11563+0.02155 |
| | HoP (−R) | −196.5510 | 0.0000 | 0.0000 | 0.00% | 0.11563+0.02090 |
| | NN-SSL | −192.2658 | 0.2079 | 0.0115 | 100.00% | 0.00994 |
| | NN-SL | −163.0384 | 0.0000 | 0.0000 | 0.00% | 0.00985 |
| | NN-SSL-SC | −175.1464 | 0.1758 | 0.0001 | 22.18% | 0.01024 |
| | NN-SL-SC | −148.6267 | 0.0000 | 0.0000 | 0.00% | **0.00999** |
| | DC3 | −180.7169 | 0.0000 | 0.0000 | 0.00% | 0.01898 |
| | HomeoProj | – | – | – | – | – |
| **QoS-MISO WSR problem** | Optimizer | −1.3791 | 0.0000 | 0.0000 | 0.00% | 11.31842 |
| | HoP (+R) | **−1.1602** | 0.0000 | 0.0000 | 0.00% | 0.88974+0.08110 |
| | HoP (−R) | −1.1567 | 0.0000 | 0.0000 | 0.00% | **0.88974+0.07630** |
| | NN-SSL | −20.6766 | 1738.1122 | 395.4313 | 100.00% | 0.04915 |
| | NN-SL | −3.2477 | 0.1636 | 0.0559 | 100.00% | 0.04906 |
| | NN-SSL-SC | −0.4150 | 0.2886 | 0.0029 | 18.25% | 0.04989 |
| | NN-SL-SC | −0.3244 | 0.7794 | 0.0035 | 23.00% | 0.04921 |
| | DC3 | −0.3381 | 0.0000 | 0.0000 | 0.00% | 3.81508 |
| | HomeoProj | −0.3961 | 0.1622 | 0.0030 | 20.00% | 0.21089 |

- HomeoProj: This baseline follows Liang et al. (2023), sharing the same loss in DC3 with additional homeomorphic post-projection to control violation.

To ensure a fair evaluation, we use the DC3 metrics. Table 1 reports: (1) Obj. Value: $f_\mathbf{x}(\hat{\mathbf{y}})$; (2) Max./Mean Cons.: $\max \mathcal{P}(\hat{\mathbf{y}})$ and $\mathrm{mean}\, \mathcal{P}(\hat{\mathbf{y}})$; (3) Vio. Rate: percentage of predicted infeasible solutions; (4) Time: computational cost (for HoP, interior-point + HoP). In the table, bold marks the best among methods that satisfy all constraints; underline indicates methods with violations. The optimizer is excluded. Further setup details appear in Appendix B.

## 4.1 Synthetic benchmarks

To evaluate HoP, we consider three different synthetic problems: polygon-constrained, non-convex $\ell_p$-norm, high-dimensional semi-unbounded problem. The polygon-constrained problem serves as the primary validation for convergence; the $\ell_p$-norm problem, a classical sparse learning formulation Donoho et al. (2011); Yang et al. (2022), serves as a benchmark to evaluate the model's capability in handling structured non-convexity, which remains tractable under the convex HoP framework (see Appendix D); the high-dimensional semi-unbounded problem tests the method's scalability and effectiveness in handling semi-unbounded large scale constraints in higher dimensions.

**(a) Polygon constrained problem** As the first benchmark experiment, we choose sinusoidal quadratic programming (QP) as the non-convex objective function with linear constraints. The problem is formulated as follows:

$$\min_{\mathbf{y}} \quad \frac{1}{2}\mathbf{y}^T\mathbf{Q}\mathbf{y} + \mathbf{p}^T \sin(\beta\mathbf{y}), \quad \text{s.t.} \quad \mathbf{A}\mathbf{y} \leq \mathbf{b}, \tag{14}$$

where matrix $\mathbf{Q}$ is a positive semi-definite matrix, vector $\mathbf{p}$ is a parameter vector, and scalar $\beta$ controls the frequency of the sinusoidal terms, which introduces non-convexity into the objective function. The constraint are defined by matrix $\mathbf{A}$ and vector $\mathbf{b}$. In this problem, the parameter $\mathbf{x}$ is $\mathbf{b}$, consistent

with the setup in Donti et al. (2021). Table 1 provides the results for the problem equation 14 under an eight-sided polygon constraint. HoP(+R) perfectly enforces feasibility, whereas NN-based methods exhibit varying levels of constraint violations. Furthermore, HoP runs over 15× faster than the optimizer, and achieve an objective of -29.7170 which nearly matches the optimizer (-29.7252) and outperforms DC3 (-29.6893) and HomeoProj (-29.3131). Notably, the ablation baseline, HoP (−R), stagnates at the origin, whereas the objective value, -0.8133, significantly underperforming all baselines, which highlights the need for reconnection. As HoP (−R) consistently fails in subsequent experiments, we focus on the analysis on HoP (+R) which is simply referred as 'HoP' throughout the rest of the paper.

**(b) Non-convex $\ell_p$-norm problem**   The second problem, an $\ell_p$-norm problem, features a QP objective function and non-convex constraints:

$$\min_{\mathbf{y}} \quad \frac{1}{2}\mathbf{y}^T\mathbf{Q}\mathbf{y} + \mathbf{p}^T\mathbf{y}, \quad \text{s.t.} \quad ||\mathbf{y}||^p_{\ell_p} \leq b \tag{15}$$

where vector $\mathbf{p}$ serves as the input variable $\mathbf{x}$ defined in Eq. (15). While our HOP method was designed for convex constraints, it naturally extends to certain non-convex cases. The $\ell_p$-norm constraint with $p < 1$ is non-convex, yet it satisfies a key geometric property: any ray emanating from the origin intersects the constraint boundary $||\mathbf{y}||^p_{\ell_p} = b$ at exactly one point. This single-intersection property ensures that the polar parameterization remains well defined, allowing HoP to be applied without modification despite the non-convexity. As shown in Table 1, although HoP, DC3, and HomeoProj all achieve perfect constraint satisfaction (0% violation), the HoP obtains objective of -0.3886, outperforming DC3's -0.0730 and HomeoProj's -0.0345. The significant optimality gap can be attributed to the post-correction mechanisms in DC3 and HomeoProj which misalign with the objective descent direction, leading to a trade-off that fundamentally limits their ability to jointly ensure feasibility and attain optimal solutions. Especially, under complex boundaries problem like $\ell_p$-norm optimization, this post-correction exacerbate the conflict. In contrast, HoP integrates constraint handling and objective descent by explicit homeomorphic mapping. Since $\mathbf{y}_0$ is origin in $\ell_p$-norm problem, the interior point is given offline. Computationally, HoP is more than 150× faster than the traditional optimizer and matches the speed of the NN-based methods. Unlike other basic NN-based approaches which exhibit violative results and inferior objective values, HoP strictly enforces constraints while delivering superior solution quality. These results highlight HoP's effectiveness in handling hard constraints.

Beyond the $\ell_p$-norm case, HoP can be extended to handle more general non-convex constraints. For non-convex constraints where rays from the origin may intersect the boundary multiple times, we propose a systematic method that partitions the feasible region into sectors. Within each sector, the single intersection property holds locally, and these sectors are then reconstructed and connected through a piecewise-linear extension. This reconstruction process connects the valid intervals along each ray, establishing a bijective mapping between the non-convex constraint space and the unit ball. This enables the our framework is able to apply on these complex constraints. We provide a detailed exposition of this extension with a demonstrative experiment in Appendix G, this piecewise extension not only demonstrates the versatility of our approach but also proves to be remarkably effective in practical engineering applications with complex non-convex constraints.

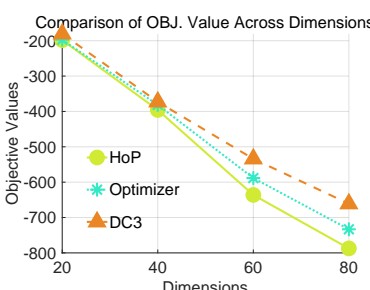

Figure 4: Comparison of objective values across methods under different dimensions.

**(c) High-dimensional semi-unbounded problem**   This experiment demonstrates HoP's ability to handle high-dimensional, semi-unbounded problems, reusing the sinusoidal QP of Eq. (14) with the input $\mathbf{p} = \mathbf{x}$. As shown in Table 1, HoP strictly satisfies all constraints and achieves a near-optimal objective of $-198.3798$, better than the optimizer's $-197.3113$, while being 226× faster. Notably, HomeoProj fails in this setting due to the divergence of its invertible NN-based homeomorphic map, which relies on the compactness of the target set. Semi-unbounded constraints break this assumption, leading to unstable training while the class of constraints is commonly encountered in practice. For example, a single hyperplane constraint can result in a semi-unbounded feasible set.

Hence, ensuring robustness under such conditions is of practical significance. Notably, NN-SL and NN-SL-SC also achieve 0 violation rates in this high-dimensional semi-unbounded experiment. This is likely due to the expanded feasible region in such scenarios, where constraints are less tight, and the optimal solution lies away from the boundary. However, it does not imply that SL-based methods can theoretically guarantee hard constraint satisfaction under all conditions.

To further evaluate the performance of different methods under higher-dimensional settings, we pushed each method to 20-, 40-, 60-, and 80-dimensional versions of the task. As shown in Fig. 4, HoP consistently performs near the optimizer in terms of objective value and exhibits a trend better than the optimizer's performance as dimensionality increases. In contrast, as the dimensionality rises, the DC3 method exhibits larger objective gap with optimizer results. These results highlight the robustness and scalability of HoP, even in high-dimensional scenarios.

### 4.2 APPLICATION – QoS-MISO WSR PROBLEM

In this experiment, we implement HoP to address the QoS-MISO WSR problem. The QoS-MISO WSR problem is a well-known NP-hard problem with non-linear constraints in communication engineering (Tang et al., 2023; Niu et al., 2021). In most studies on the QoS-MISO WSR problem, researchers commonly employ alternative optimization methods to obtain solutions, which often require significant computational resources. The problem is formulated as follows:

$$\min_{\mathbf{w}_k} -\sum_{k=1}^{U} \alpha_k \log_2(1 + \text{SINR}_k) \tag{16}$$

$$\text{s.t. } \log_2(1 + \text{SINR}_k) \geq \delta_k, \quad \sum_{k=1}^{U} \text{Tr}(\mathbf{w}_k \mathbf{w}_k^H) + \text{P}_c \leq \text{P}_{\max} \tag{16a}$$

where $\mathbf{w}_k \in \mathbb{C}^M$ is the beamformer, $(.)^H$ represents Hermitian transpose, $\mathbf{h}_k \in \mathbb{C}^M$ denotes channel state information, and $\sigma_k^2 \in \mathbb{R}$ is channel noise power. $\alpha_k \in \mathbb{R}$ denotes priority weight for $U$ users, $\delta_k$ represents the QoS requirements. $\text{P}_{\max}$ and $\text{P}_c$ define the system maximum power and circuit power consumption, respectively. The signal-interference-noise-ratio $\text{SINR}_k$ is defined as

$$\text{SINR}_k = \frac{\mathbf{w}_k^H \mathbf{h}_k \mathbf{h}_k^H \mathbf{w}_k}{\sum_{j \neq k}^{U} \mathbf{w}_j^H \mathbf{h}_k \mathbf{h}_k^H \mathbf{w}_j + \sigma_k^2} \tag{17}$$

Given that wireless resource allocation requires real-time optimization strategies to effectively manage wireless resources with diverse channel state information, we select $\mathbf{h}_k$ as the input of NN. This selection is crucial as $\mathbf{h}_k$ significantly influences both the constraints and the objective function in Eq. (16). For a comprehensive understanding of how we apply HoP to the QoS-MISO WSR problem, please refer to Appendix B.2. According to Table 1, both HoP and DC3 achieve perfect constraint satisfaction, with $0\%$ violation rate. However, due to the non-stationary constraint boundaries in QoS-MISO WSR problem, the validity of HomoeProj no longer satisfied the sufficient conditions given in Liang et al. (2023) which cause violation samples in Table 1. NN-based methods are consistent with previous experiments, struggling with constraint satisfaction. Especially, NN-SSL and NN-SL have extremely high Max, Cons and a Mean. Cons, with $100\%$ violation rate, showing that it fails to guarantee feasibility. For computation efficiency, HoP is slower than basic NN solvers, while it still gains a huge speedup, 11 times faster than the traditional algorithm. While the Optimizer (SCS+FP) achieves the best objective value, HoP obtains a competitive result, which outperforms other NN solvers while maintaining perfect constraint satisfaction. Thus, HoP is a highly effective and reliable method for solving the problem in engineering such as the QoS-MISO WSR problem.

## 5 CONCLUSION

In this work, we propose HoP, a novel L2O framework for solving hard-constrained optimization problems. The proposed architecture integrates NN predictions with a homeomorphic mapping, which transforms NN's outputs from spherical polar space to Cartesian coordinates, ensuring solution feasibility without extra penalties or post-correction. Through extensive experiments encompassing both synthetic benchmark tasks and real-world applications, we demonstrate that HoP consistently outperforms existing L2O solvers, achieving superior optimality while maintaining zero constraint violation rates.

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

## A  JACOBIAN ANALYSIS AND MEASURE DISTORTION

This section presents a detailed analysis of the transformation

$$\hat{\mathbf{y}}(\psi, \mathbf{v}_\theta) \;=\; \mathbf{y}_0 + \mathbf{v}_\theta \tan(\psi), \tag{18}$$

where $\psi \in [0, \frac{\pi}{2})$ is a scalar parameter, and $\mathbf{v}_\theta \in \mathbb{R}^d$ is a unit vector (i.e., $\|\mathbf{v}_\theta\| = 1$). In this formulation, the direction $\mathbf{v}_\theta$ is treated as a free variable on the unit sphere $S^{d-1}$, while $\psi$ governs

the radial displacement via the function $\tan(\psi)$. The mapping in Eq. equation 18 corresponds to the higher-dimensional homeomorphic mapping introduced in Eq. equation 8. It provides a diffeomorphic embedding of the parameter space $\mathcal{P}$ into $\mathbb{R}^d$, where $\mathcal{P}$ is defined as:

$$\mathcal{P} = \left\{ (\psi, \mathbf{v}_\theta) \mid \psi \in [0, \tfrac{\pi}{2}),\ \mathbf{v}_\theta \in S^{d-1} \right\},$$

The analysis below derives the Jacobian determinant of this transformation and discusses the measure distortion that arises as $\psi$ approaches $\frac{\pi}{2}$, where $\tan(\psi)$ diverges. These results provide insight into the geometric and numerical properties of the homeomorphic mapping when applied to optimization tasks in semi-unbounded domains.

## A.1  JACOBIAN DETERMINANT

To characterize local volume distortion, we compute the Jacobian determinant (Spivak, 2018). The mapping $\hat{\mathbf{y}}(\psi, \mathbf{v}_\theta)$ is defined in Eq. equation 18. The total derivative $\mathrm{D}\hat{\mathbf{y}}(\psi, \mathbf{v}_\theta)$ is a $d \times d$ matrix whose columns represent the partial derivatives of $\hat{\mathbf{y}}$ with respect to $\psi$ and to the $(d-1)$ degrees of freedom on the unit sphere $S^{d-1}$. Specifically:

**(a) Derivative w.r.t. $\psi$.**  For fixed $\mathbf{v}_\theta$,

$$\frac{\partial}{\partial \psi} \big( \tan(\psi)\, \mathbf{v}_\theta \big) = \sec^2(\psi)\, \mathbf{v}_\theta. \tag{19}$$

**(b) Derivatives w.r.t. the sphere parameters $\mathbf{v}_\theta$.**

**Lemma A.1** (Tangent Space). *Let $\mathbf{v}_\theta \in S^{d-1}$ be a unit vector on the sphere in $\mathbb{R}^d$, satisfying $\|\mathbf{v}_\theta\| = 1$. Then any infinitesimal variation $\mathrm{d}\mathbf{v}_\theta$ must lie in the tangent space $T_{\mathbf{v}_\theta} S^{d-1}$, given by:*

$$\mathbf{v}_\theta \cdot \frac{\mathrm{d}\mathbf{v}_\theta}{\mathrm{d}\theta} = 0 \quad \implies \quad \mathrm{d}\mathbf{v}_\theta \in T_{\mathbf{v}_\theta} S^{d-1}.$$

*The tangent space $T_{\mathbf{v}_\theta} S^{d-1}$ is a $(d-1)$-dimensional subspace of $\mathbb{R}^d$.*

*Proof.* The constraint $\|\mathbf{v}_\theta\| = 1$ implies $\mathbf{v}_\theta^T \cdot \mathbf{v}_\theta = 1$. Differentiating this equation with respect to $\theta$ yields:

$$\frac{\mathrm{d}}{\mathrm{d}\theta}\big( \mathbf{v}_\theta^T \cdot \mathbf{v}_\theta \big) = 2\, \mathbf{v}_\theta \cdot \frac{\mathrm{d}\mathbf{v}_\theta}{\mathrm{d}\theta} = 0.$$

Thus, any allowed variation $\frac{\mathrm{d}\mathbf{v}_\theta}{\mathrm{d}\theta}$ is orthogonal to $\mathbf{v}_\theta$, and therefore lies in the tangent space $T_{\mathbf{v}_\theta} S^{d-1}$, which is the subspace of $\mathbb{R}^d$ orthogonal to $\mathbf{v}_\theta$.

Since $S^{d-1}$ is a $(d-1)$-dimensional manifold embedded in $\mathbb{R}^d$, its tangent space $T_{\mathbf{v}_\theta} S^{d-1}$ is also $(d-1)$-dimensional. $\qquad\square$

**Lemma A.2** (Linear Independence of $\{\mathbf{v}_\theta, \mathbf{w}_1, \ldots, \mathbf{w}_{d-1}\}$). *Let $\mathbf{v}_\theta \in S^{d-1}$ be a unit vector, and let $\{\mathbf{w}_1, \ldots, \mathbf{w}_{d-1}\}$ be an orthonormal basis of $T_{\mathbf{v}_\theta} S^{d-1}$, satisfying:*

$$\langle \mathbf{v}_\theta, \mathbf{w}_i \rangle = 0, \quad \langle \mathbf{w}_i, \mathbf{w}_j \rangle = \begin{cases} 1, & \text{if } i = j, \\ 0, & \text{if } i \neq j, \end{cases} \quad i, j = 1, \ldots, d-1.$$

*Then the set $\{\mathbf{v}_\theta, \mathbf{w}_1, \ldots, \mathbf{w}_{d-1}\}$ is linearly independent in $\mathbb{R}^d$. Specifically, any linear combination*

$$a\, \mathbf{v}_\theta + \sum_{i=1}^{d-1} b_i\, \mathbf{w}_i = \mathbf{0}$$

*implies $a = 0$ and $b_i = 0$ for all $i$.*

*Proof.* Assume the linear dependence:

$$a\, \mathbf{v}_\theta + \sum_{i=1}^{d-1} b_i\, \mathbf{w}_i = \mathbf{0}.$$

Taking the inner product with $\mathbf{v}_\theta$, we obtain:

$$a \langle \mathbf{v}_\theta, \mathbf{v}_\theta \rangle + \sum_{i=1}^{d-1} b_i \langle \mathbf{w}_i, \mathbf{v}_\theta \rangle = a \|\mathbf{v}_\theta\|^2 = a = 0,$$

since $\|\mathbf{v}_\theta\| = 1$ and $\mathbf{v}_\theta \cdot \mathbf{w}_i = 0$. Next, taking the inner product with $\mathbf{w}_j$, we have:

$$\sum_{i=1}^{d-1} b_i \langle \mathbf{w}_i, \mathbf{w}_j \rangle = b_j,$$

because $\langle \mathbf{w}_i, \mathbf{w}_j \rangle = 0$ for $i \neq j$ and $\langle \mathbf{w}_j, \mathbf{w}_j \rangle = 1$. Hence, $b_j = 0$.

Since $j$ is arbitrary, we conclude that $b_i = 0$ for all $i$, and thus $a = 0$. This proves that the set $\{\mathbf{v}_\theta, \mathbf{w}_1, \ldots, \mathbf{w}_{d-1}\}$ is linearly independent. $\qquad\square$

Given the mapping $\hat{\mathbf{y}}(\psi, \mathbf{v}_\theta) = \tan(\psi)\,\mathbf{v}_\theta$, any infinitesimal change of $\mathbf{v}_\theta$ on the unit sphere $S^{d-1}$ lies in the tangent space $T_{\mathbf{v}_\theta} S^{d-1}$ (see Lemma A.1), ensuring $\mathbf{v}_\theta \cdot d\mathbf{v}_\theta = 0$. We introduce an orthonormal basis $\{\mathbf{w}_1, \ldots, \mathbf{w}_{d-1}\}$ for this $(d-1)$-dimensional space. Consequently, each partial derivative with respect to $\mathbf{v}_\theta$ appears in the direction of some $\mathbf{w}_i$, yielding

$$\frac{\partial \hat{\mathbf{y}}}{\partial v_{\theta,i}} = \tan(\psi)\,\mathbf{w}_i, \quad i = 1, \ldots, d-1.$$

Hence, $\hat{\mathbf{y}}$ increases linearly in each $\mathbf{w}_i$-direction, with a scaling factor $\tan(\psi)$.

Collecting all these derivatives, The Jacobian matrix of the transformation is given by:

$$\mathrm{D}\hat{\mathbf{y}}(\Phi, \mathbf{v}_\theta) = \begin{bmatrix} \sec^2(\psi)\,v_{\theta,1} & \tan(\psi)\,w_{1,1} & \ldots & \tan(\psi)\,w_{d-1,1} \\ \sec^2(\psi)\,v_{\theta,2} & \tan(\psi)\,w_{1,2} & \ldots & \tan(\psi)\,w_{d-1,2} \\ \vdots & \vdots & \ddots & \vdots \\ \sec^2(\psi)\,v_{\theta,d} & \tan(\psi)\,w_{1,d} & \ldots & \tan(\psi)\,w_{d-1,d} \end{bmatrix}.$$

Here, the first column corresponds to $\frac{\partial \mathbf{x}}{\partial \Phi}$, and the subsequent $(d-1)$ columns represent $\frac{\partial \mathbf{x}}{\partial v_i}$ for each tangent direction $\mathbf{w}_i$.

Since $\{\mathbf{v}_\theta, \mathbf{w}_1, \ldots, \mathbf{w}_{d-1}\}$ is an orthonormal set (as established in Lemma A.2, with $\langle \mathbf{v}_\theta, \mathbf{w}_i \rangle = 0$, $\|\mathbf{v}_\theta\| = 1$, and $\|\mathbf{w}_i\| = 1$), the determinant of $\mathrm{D}\hat{\mathbf{y}}(\psi, \mathbf{v}_\theta)$ equals the product of the column norms:

$$\det\big(\mathrm{D}\hat{\mathbf{y}}(\psi, \mathbf{v}_\theta)\big) = \big\|\sec^2(\psi)\,\mathbf{v}_\theta\big\| \times \prod_{i=1}^{d-1} \big\|\tan(\psi)\,\mathbf{w}_i\big\|.$$

Noting that $\|\mathbf{v}_\theta\| = 1$ and $\|\mathbf{w}_i\| = 1$, the norms are

$$\big\|\sec^2(\psi)\,\mathbf{v}_\theta\big\| = \sec^2(\psi), \quad \big\|\tan(\psi)\,\mathbf{w}_i\big\| = \tan(\psi).$$

Hence, the determinant simplifies to

$$\det\big(\mathrm{D}\hat{\mathbf{y}}(\Phi, \mathbf{v}_\theta)\big) = \big(\sec^2(\psi)\big) \times \big(\tan(\psi)\big)^{d-1} = \tan(\psi)^{d-1} \sec^2(\psi).$$

### A.2 MEASURE DISTORTION NEAR $\psi \to \pi/2$

**Theorem A.3** (Regularization of Jacobian Divergence). *Let $\hat{\mathbf{y}}(\psi, \mathbf{v}_\theta) = \tan(\psi)\,\mathbf{v}_\theta$ be the homeomorphic mapping defined in HoP, where $\psi \in (0, \frac{\pi}{2})$ and $\mathbf{v}_\theta \in S^{d-1}$. The Jacobian determinant of $\hat{\mathbf{y}}$ is given by:*

$$\det\big(\mathrm{D}\hat{\mathbf{y}}(\psi, \mathbf{v}_\theta)\big) = \tan(\psi)^{d-1} \sec^2(\psi).$$

*As $\psi \to \frac{\pi}{2}$, the Jacobian determinant diverges as:*

$$\det\big(\mathrm{D}\hat{\mathbf{y}}(\psi, \mathbf{v}_\theta)\big) \sim \frac{1}{\epsilon^{d+1}},$$

*where $\epsilon = \frac{\pi}{2} - \psi$. To address this divergence, HoP introduces $\epsilon > 0$ as a regularization parameter, ensuring that:*

1. *The mapping $\hat{\mathbf{y}}$ remains well-defined and smooth for all $\psi \in (-\frac{\pi}{2}, \frac{\pi}{2} - \epsilon]$.*

2. *The Jacobian determinant is bounded by:*

$$\det\big(\mathrm{D}\hat{\mathbf{y}}(\psi, \mathbf{v}_\theta)\big) \leq C \cdot \frac{1}{\epsilon^{d+1}},$$

   *where $C$ is a constant depending only on $d$.*

*This regularization prevents unbounded volume distortion and ensures numerical stability in optimization and sampling procedures.*

*Proof.* The divergence follows from the asymptotic behavior of $\tan(\psi)$ and $\sec(\psi)$ as $\psi \to \frac{\pi}{2}$:

$$\tan(\psi) \sim \frac{1}{\epsilon} \quad \text{and} \quad \sec^2(\psi) \sim \frac{1}{\epsilon^2}.$$

Substituting these into the Jacobian determinant yields:

$$\det\big(\mathrm{D}\hat{\mathbf{y}}(\psi, \mathbf{v}_\theta)\big) = \tan(\psi)^{d-1} \sec^2(\psi) \sim \frac{1}{\epsilon^{d-1}} \cdot \frac{1}{\epsilon^2} = \frac{1}{\epsilon^{d+1}}.$$

By restricting $\psi$ to $\psi \leq \frac{\pi}{2} - \epsilon$, the Jacobian determinant remains bounded, completing the proof. $\square$

Thus, the Jacobian determinant diverges polynomially in $\frac{1}{\epsilon}$, reflecting a severe measure distortion in the limit $\psi \to \frac{\pi}{2}$. In practical terms, small parameter increments around $\psi \approx \frac{\pi}{2}$ map to disproportionately large volume elements in $\mathcal{Y}_\mathbf{x}$, leading to potential numerical instability if one attempts to sample or optimize directly over $\psi$ without truncation.

Although the Jacobian determinant $\det\big(\mathrm{D}\hat{\mathbf{y}}(\psi, \mathbf{v}_\theta)\big)$ diverges as $\psi \to \frac{\pi}{2}$, practical considerations ensure numerical stability. Empirical knowledge or problem-specific constraints often imply an upper bound $\psi_{\max} < \frac{\pi}{2}$, allowing the selection of $\epsilon = \frac{\pi}{2} - \psi_{\max}$. Additionally, finite precision in computing hardware inherently limits the representable range, preventing true divergence in practice.

# B EXPERIMENTAL DATA GENERATION AND PARAMETER SETUP

In this section, the hyper-parameters of pervious experiments are provided. We split the section into two subsections to introduce the setting in Experiment 4.1 and Experiment 4.2, respectively.

All experiments were conducted on a system with the following specifications:

- Host: Lambda Vector 1
- OS: Ubuntu 22.04.5 LTS x86_64
- CPU: AMD Ryzen Threadripper PRO 5955WX (32 cores) @ 4.000GHz
- GPU: Single NVIDIA RTX A6000

To simulate full parallelization, we report the total computation time divided by the number of test instances. Note that our implementations are not tightly optimized, and all timing comparisons should be interpreted as approximate.

## B.1 SYNTHETIC BENCHMARKS 4.1

### B.1.1 (A) POLYGON-CONSTRAINED PROBLEM

The objective function for this experiment is defined as in Eq. (14). The problem is set in a 2-dimensional space, with the matrix $\mathbf{Q}$ randomly generated as a positive semi-definite matrix to ensure convex quadratic terms. Specifically, $\mathbf{Q}$ is created using the following procedure:

$$\mathbf{Q} = \mathbf{A}^T\mathbf{A} + \mathbf{I}, \tag{20}$$

where $\mathbf{A}$ is a randomly generated $2 \times 2$ matrix drawn from a standard Gaussian distribution, and $\mathbf{I}$ is the identity matrix to guarantee positive semi-definiteness.

The vector $\mathbf{b}$ is generated by sampling from a uniform distribution $\mathcal{U}(0,2)$ and is checked against the polygon constraints using a validation function. Specifically, $\mathbf{b}$ is accepted only if it satisfies the closure conditions of the polygon constraints. Samples violating these constraints are discarded, ensuring all 20,000 instances in the dataset strictly satisfy the feasibility conditions.

The dataset is divided into a training set (70%) and a test set (30%). The parameters in the objective function are set as follows: the vector $\mathbf{p}$ is fixed to $[30, 30]$, and the sinusoidal term's parameters $\mathbf{q}$ are randomly generated for each instance, also drawn from $\mathcal{U}(0,2)$. The constraint matrix $\mathbf{A}$ is precomputed and fixed for all experiments.

The center point $\mathbf{y}_0$ of the polygon is computed as the Chebyshev center of the feasible region. The Chebyshev center is obtained by solving a secondary optimization problem that maximizes the radius of the largest inscribed circle within the polygon constraints, ensuring that $\mathbf{y}_0$ lies strictly inside the feasible region.

### B.1.2 (B) $\ell_p$-NORM PROBLEM

The objective function for this experiment is defined in Eq. (15). The matrix $\mathbf{Q}$ is randomly generated as a positive semi-definite matrix, ensuring convex quadratic terms, while the vector $\mathbf{p}$ is sampled from a standard normal distribution $\mathcal{N}(0,1)$. The $\ell_p$-norm constraint uses $p = 0.5$, and the parameter $b$ is fixed to 1. The dataset consists of 20,000 samples, split into a training set (70%) and a test set (30%). The center point $\mathbf{y}_0$ is set to $[0, 0]$.

The positive semi-definite matrix $\mathbf{Q}$ is generated as described in Eq. (20).

The dataset is constructed by generating feasible samples that satisfy the $\ell_p$-norm constraint. Instances violating the constraint are discarded, ensuring that the entire dataset strictly adheres to the defined feasibility conditions.

### B.1.3 (C) HIGH-DIMENSIONAL SEMI-UNBOUNDED PROBLEM

The objective function for high-dimensional semi-unbounded problem is given in Eq. (14), where $\beta = 30$, and $\mathbf{p} \sim \mathcal{N}$ is randomly sampled from the Gaussian Distribution $(-10, \mathbf{I})$. The parameters in the constraints are fixed, where $\mathbf{A}$ is randomly drawn from $\mathcal{N}(0, \mathbf{I})$ and $\mathbf{b}$ is drawn randomly as a positive number. Since in this experiment constraints are fixed for a specific dataset, without lose of generality we define $\mathbf{b} = \mathbf{1}$. Moreover, the number of linear constraints is $d$, which is the dimension of $\hat{\mathbf{y}}$. Thus arbitrary semi-unbounded constraints are obtained while we have $\mathbf{0}$ as $\mathbf{y}_0$. The dataset is divided into a training set (70%) and a test set (30%). For training part, the learn rate is $10^{-4}$, the optimizer is Adam, and the training configuration includes 500 epochs with a batch size of $2,048$. For 20-dimensional problem, the dataset contains $20,000$ instance; 40, 60, 80 dimensions problems have $40,000, 60,000, 80,000$ instances, respectively. The MLP used for both all baseline methods and HoP has 512 neurons with ReLU activation functions.

### B.1.4 (D) NON-CONVEX PARABOLIC CONSTRAINED PROBLEM

The objective function used for the high-dimensional semi-unbounded experiment follows Eq. (14), where the scaling parameter is set to $\beta = 30$. The matrix $\mathbf{Q}$ in the quadratic term is randomly generated as a positive semi-definite matrix to ensure convexity, while the linear coefficient vector $\mathbf{p}$ is sampled from a standard normal distribution, i.e., $\mathbf{p} \sim \mathcal{N}(0, \mathbf{I})$.

The construction of the positive semi-definite matrix $\mathbf{Q}$ adheres to the method described in Eq. (20). Similarly, each constraint matrix $\mathbf{A}'_i$ used in the parabolic-constrained is also constructed using Eq. (20). However, to address the issue that parabolic surfaces tend to become excessively sharp or narrow in higher-dimensional settings—making the feasible region overly restrictive—we manually reduce the curvature by scaling the generated matrix with a factor of $10^{-5}$. This adjustment ensures better numerical stability and smoother constraint surfaces in high-dimensional space.

The vector $\mathbf{b}_i$ is independently sampled from a standard normal distribution, analogous to $\mathbf{p}$. The scalar constants $c_i$ are randomly drawn from a uniform distribution over the interval $[0.1, 0.2]$ to provide moderate vertical offsets in the constraint surfaces.

The experiments are conducted in $\mathbb{R}^{20}$, i.e., the output dimension is set to 20.

## B.2 QoS-MISO WSR PROBLEM

In this subsection the data preparation and how we apply HoP on QoS-MISO WSR problem are demonstrated. To generate the MISO simulation data, we apply algorithm given in Appendix C. The user priority weight $\alpha_k \sim \mathcal{U}(0,1)$ with uniformization by $\alpha_k/(\sum_{k=1}^{U} \alpha_k)$. Channel state information is $\mathbf{h}_k$, followed circularly symmetric complex Gaussian (CSCG) distribution where the real part and imagine part follows $\mathcal{CN}(0,1)$. pathloss $= 10$ and $\sigma^2 = 0.01$, $\delta_k \sim \mathcal{U}(0, 1/3)$. $P_{max} = 33$ dbm, $P_c = 30$ dbm. For the training part, the learn rate is $10^{-2}$, the optimizer is Adam, and the training configuration includes $500$ epochs with a batch size of $64$. The dataset has $4000$ instances which is divided into a training set ($70\%$) and a test set ($30\%$). Then, in the following part, we introduce how to formulate this multi-variables problem as a single variable problem which satisfies the NN's outputs format. The problem given in Eq. (16) can be reformulated as:

$$\max_{\mathbf{w}_k} \sum_{k=1}^{U} \alpha_k \log_2(1 + \text{SINR}_k) \tag{21}$$

$$\text{s.t.} \quad \frac{\mathbf{w}_k^H \mathbf{h}_k \mathbf{h}_k^H \mathbf{w}_k}{\sum_{j \neq k}^{U} \mathbf{w}_j^H \mathbf{h}_k \mathbf{h}_k^H \mathbf{w}_j + \sigma_k^2} \geq \omega_k \tag{21a}$$

$$\sum_{k=1}^{U} P_k \leq P_{max} - P_c \tag{21b}$$

where $\omega_k = 2^{\delta_k} - 1$. Then, the constraint in Eq. (21a) is identical to:

$$\mathbf{w}_k^H \mathbf{h}_k \mathbf{h}_k^H \mathbf{w}_k - \omega_k \sum_{j \neq k}^{U} \mathbf{w}_j^H \mathbf{h}_k \mathbf{h}_k^H \mathbf{w}_j \geq \omega_k \sigma_k^2 \tag{22}$$

In this problem, for the convenience, $\mathbf{w}_k$ and $\mathbf{h}_k$ are reformulated as real vector and matrix by splicing:

$$\tilde{\mathbf{H}}_k = \begin{bmatrix} \Re(\mathbf{h}_k \mathbf{h}_k^H) & -\Im(\mathbf{h}_k \mathbf{h}_k^H) \\ \Im(\mathbf{h}_k \mathbf{h}_k^H) & \Re(\mathbf{h}_k \mathbf{h}_k^H) \end{bmatrix} \tag{23}$$

$$\tilde{\mathbf{w}}_k = \begin{bmatrix} \Re(\mathbf{w}_k) \\ \Im(\mathbf{w}_k) \end{bmatrix} \tag{24}$$

Therefore, Eq. (22) is represented by Eq. (23) and Eq. (24):

$$\tilde{\mathbf{w}}_k^T \tilde{\mathbf{H}}_k \tilde{\mathbf{w}}_k - \omega_k \sum_{j \neq k}^{U} \tilde{\mathbf{w}}_j^T \tilde{\mathbf{H}}_k \tilde{\mathbf{w}}_j \geq \omega_k \sigma_k^2 \tag{25}$$

Splicing all $\tilde{\mathbf{w}}_k$ for each user, then Eq. (25) is rewritten as:

$$\bar{\mathbf{w}}^T \bar{\mathbf{H}}_k \bar{\mathbf{w}} \geq \omega_k \sigma_k^2 \tag{26}$$

where $\bar{\mathbf{w}}$, $\bar{\mathbf{H}}_k$ and $f_j(\omega_k)$ are defined as:

$$\bar{\mathbf{w}} = [\tilde{\mathbf{w}}_1^T, \tilde{\mathbf{w}}_2^T, ..., \tilde{\mathbf{w}}_U^T]^T \tag{27}$$

$$\bar{\mathbf{H}}_k = \begin{bmatrix} f_1(\omega_k)\tilde{\mathbf{H}}_k & \mathbf{0} & ... & \mathbf{0} \\ \mathbf{0} & f_2(\omega_k)\tilde{\mathbf{H}}_k & ... & \mathbf{0} \\ \vdots & \vdots & \ddots & \vdots \\ \mathbf{0} & \mathbf{0} & ... & f_U(\omega_k)\tilde{\mathbf{H}}_k \end{bmatrix} \tag{28}$$

$$f_j(\omega_k) \begin{cases} 1, & j = k, \\ -\omega_k, & j \neq k. \end{cases} \tag{29}$$

Based on above transformation, we apply the same operation on other constraints, then we have

$$\bar{\mathbf{w}}^T \bar{\mathbf{H}}_1 \bar{\mathbf{w}} \geq \omega_1 \sigma_1^2$$

$$\bar{\mathbf{w}}^T \bar{\mathbf{H}}_2 \bar{\mathbf{w}} \geq \omega_2 \sigma_2^2$$

$$\vdots$$

$$\bar{\mathbf{w}}^T \bar{\mathbf{H}}_U \bar{\mathbf{w}} \geq \omega_k \sigma_U^2 \tag{30}$$

$$\bar{\mathbf{w}}^T \mathbf{I} \bar{\mathbf{w}} \leq \mathrm{P}_{\max} - \mathrm{P}_{\mathrm{c}} \tag{31}$$

Here, the original problem can be reorganized as an equivalent problem as follows:

$$\max_{\bar{\mathbf{w}}_k} \sum_{k=1}^{U} \alpha_k \log_2(1 + \mathrm{SINR}_k) \tag{32}$$

$$\text{s.t. } \bar{\mathbf{w}}^T \bar{\mathbf{H}}_1 \bar{\mathbf{w}} \geq \omega_1 \sigma_1^2$$

$$\bar{\mathbf{w}}^T \bar{\mathbf{H}}_2 \bar{\mathbf{w}} \geq \omega_2 \sigma_2^2$$

$$\vdots$$

$$\bar{\mathbf{w}}^T \bar{\mathbf{H}}_U \bar{\mathbf{w}} \geq \omega_U \sigma_U^2 \tag{32a}$$

$$\bar{\mathbf{w}}^T \mathbf{I} \bar{\mathbf{w}} \leq \mathrm{P}_{\max} - \mathrm{P}_{\mathrm{c}} \tag{32b}$$

As consequence, $\bar{\mathbf{w}}$ is the estimated variable $\hat{\mathbf{y}}$ by HoP, where the NN's inputs, $\mathbf{x}$ are flatten $\tilde{\mathbf{H}}_k$.

## C    FP FOR QOS-MISO WSR EXPERIMENT

We use the FP method to solve the original QoS-MISO WSR problem to compute the reference optimal beamformers results. According to Shen & Yu (2018), we reformulate the original problem as

$$\max_{\mathbf{W}_k} \sum_{k=1}^{U} \alpha_k \log_2 \left( 1 + 2l_k \sqrt{\mathbf{h}_k^H \mathbf{W}_k \mathbf{h}_k} - l_k^2 (\sigma_k^2 + \sum_{j \neq k}^{U} (\mathbf{h}_k^H \mathbf{W}_j \mathbf{h}_k)) \right) \tag{33}$$

$$\text{s.t. } \log_2(1 + \frac{\mathbf{h}_k^H \mathbf{W}_k \mathbf{h}_k}{\sigma_k^2 + \sum_{j \neq k}^{U} (\mathbf{h}_k^H \mathbf{W}_j \mathbf{h}_k)}) \geq \delta_k \tag{33a}$$

$$\sum_{k=1}^{U} \mathrm{Tr}(\mathbf{W}_k) \leq \mathrm{P}_{\max} - \mathrm{P}_{\mathrm{c}} \tag{33b}$$

where $l_k$ is auxiliary variable can be obtained by:

$$l_k = \frac{\sqrt{\mathbf{h}_k^H \mathbf{W}_k \mathbf{h}_k}}{\sigma_k^2 + \sum_{j \neq k}^{U} \mathbf{h}_k^H \mathbf{W}_j \mathbf{h}_k} \tag{34}$$

Note that the variables are applied with semidefinite relaxation such that $\mathbf{W}_k = \mathbf{w}_k \mathbf{w}_k^H$, $\mathrm{rank}(\mathbf{W}_k) = 1$. Then, $\mathbf{w}_k$ is obtained by applying singular value decomposition (SVD) on $\mathbf{W}_k$. Therefore, the FP algorithm can solve QoS-MISO WSR problem by Algorithm 2.

## D    UNIQUENESS OF RAY INTERSECTION WITH $\ell_p$ BALL BOUNDARY

Let $\mathbf{r} \in \mathbb{R}^n$ be a nonzero vector, and define the ray from the origin as $\mathbf{x}(t) = t\mathbf{r}$ for $t \geq 0$. Consider the $\ell_p$-norm ball of radius $b > 0$ centered at the origin:

$$\mathcal{C} = \{\mathbf{x} \in \mathbb{R}^n \mid \|\mathbf{x}\|_p \leq b\} .$$

Then the ray intersects the boundary of $\mathcal{C}$, i.e., the set $\partial\mathcal{C} = \{\mathbf{x} \in \mathbb{R}^n \mid \|\mathbf{x}\|_p = b\}$, at a unique point.

---

**Algorithm 2** FP for QoS-MISO WSR optimization

---

**Input:** Initialization parameters. Set counter $j = 1$ and convergence precision $\varphi_p$,
**repeat**
    Update $s_k$ by (34);
    Solve (33) by cvxpy;
    Solve $\mathbf{w}_k$ by SVD;
**until** $|\text{WSR}_{j+1} - \text{WSR}_j| \leq \varphi_p$

---

**Proof.**   Let $\mathbf{x}(t) = t\mathbf{r}$ for $t \geq 0$. Then:

$$\|\mathbf{x}(t)\|_p = \|t\mathbf{r}\|_p = t \cdot \|\mathbf{r}\|_p.$$

To find the intersection with the boundary, solve:

$$\|\mathbf{x}(t)\|_p = b \quad \iff \quad t = \frac{b}{\|\mathbf{r}\|_p}.$$

Thus, the unique intersection point is:

$$\mathbf{x}^* = \frac{b}{\|\mathbf{r}\|_p} \cdot \mathbf{r}.$$

Moreover,

- For $t < \frac{b}{\|\mathbf{r}\|_p}$, we have $\|\mathbf{x}(t)\|_p < b$, so $\mathbf{x}(t) \in \text{int}(\mathcal{C})$;

- For $t > \frac{b}{\|\mathbf{r}\|_p}$, we have $\|\mathbf{x}(t)\|_p > b$, so $\mathbf{x}(t) \notin \mathcal{C}$.

Hence, the ray intersects the boundary at exactly one point:

$$\mathbf{x}^* = \frac{b}{\|\mathbf{r}\|_p} \cdot \mathbf{r}.$$

∎

## E   IMPLEMENTATION PERFORMANCE IMPROVEMENT DETAILS

In this section, the implementation performance details are provided which include the approach to solve redundant constraints problem in convex case and non-convex case, the numeric computing acceleration for large scale constraints.

### E.1   REDUNDANT CONSTRAINTS

The redundant constraints have two distinct categories: convex and non-convex constraints.

**Redundant constraints in convex set**   In the convex case, each ray originating from the polar center $\mathbf{y}_0$ intersects the boundary of the feasible set exactly once along any given direction $\mathbf{v}_\theta$. This implies that the farthest intersection point, denoted by $s_1 = \mathbf{y}_0 + R(\mathbf{v}_\theta, \mathcal{C})\mathbf{v}_\theta$, is sufficient to define the feasible boundary in that direction. Consequently, any constraint that does not contribute to defining this boundary is automatically rendered redundant, and the formulation remains unaffected by their presence. The proof has been provided in Appendix F.

**Redundant constraints in non-convex set**   In contrast, for non-convex constraint sets, a ray may intersect the boundary multiple times due to the disconnected or folded structure of the feasible region. To systematically identify and eliminate redundant constraints under such conditions, we begin by computing all intersection points between the ray $\mathbf{y}_0 + t\mathbf{v}_\theta$ and the constraint boundaries. Each intersection point is then substituted back into the full constraint set to verify whether any constraints are violated at that location. If a point lies outside some constraint boundary, this indicates that the corresponding constraint is non-redundant in that direction. If the point satisfies all constraints, but does not correspond to a true boundary transition, it is recorded but treated accordingly.

After collecting all valid intersections, we sort them by their distance to $\mathbf{y}_0$ and denote the ordered points as $s_1, s_2, \ldots, s_N$. According to the Jordan curve theorem, the segments between each pair of intersection points alternate between feasible and infeasible regions. Therefore, the intervals from $s_{2t}$ to $s_{2t+1}$ correspond to feasible segments, which we refer to as effective segments. These segments define the valid solution region along the ray, and their corresponding angular spans form the effective angle domains used in our homeomorphic mapping framework, as described in the main body of the paper.

### E.2 ACCELERATED BOUNDARY COMPUTATION VIA NUMERICAL METHODS

To improve the computational efficiency of locating these boundary points, we adopt numerical techniques tailored to the geometry of the feasible set.

In the convex case, the feasibility boundary along a given direction is unique and well-defined, which allows for an efficient binary search strategy. We initialize the search interval with the interior point $\mathbf{y}_0$, corresponding to zero distance, and a sufficiently large point along $\mathbf{v}_\theta$ that is known to lie outside the feasible region. By evaluating feasibility at the midpoint of this interval and recursively narrowing the range, the algorithm converges rapidly to the exact boundary point. This method is both stable and accurate under convex assumptions.

However, in the non-convex case, binary search alone is insufficient due to the presence of multiple feasible segments along a single ray. Instead, we perform dense sampling along the ray direction and monitor changes in constraint satisfaction to detect potential boundary transitions. Whenever the sign of a constraint function changes or an activation pattern shifts, we record the corresponding position as an approximate intersection point. These candidate points are then used in constructing the effective segments, without explicitly relying on a closed-form or iterative solver. Although less efficient than the convex case, this method preserves generality and enables robust support for arbitrary non-convex geometries.

By combining this numerical boundary identification approach with the theoretical structure of Jordan segmentation, we ensure that only the necessary and active constraints participate in defining the feasible region. This not only improves computational speed but also eliminates redundant constraints dynamically, enabling efficient and accurate constrained optimization.

## F REDUNDANT CONSTRAINTS IN CONVEX CASE

*Proof.* Since $C$ is convex with respect to $\mathbf{y}_0$, for any point $\mathbf{y} \in C$, the line segment connecting $\mathbf{y}_0$ and $\mathbf{y}$ is entirely contained in $C$. That is, for all $t \in [0, 1]$, we have:

$$(1 - t)\mathbf{y}_0 + t\mathbf{y} \in C. \tag{35}$$

Then we consider a ray originating from $\mathbf{y}_0$ in the direction of a unit vector $\mathbf{v}_\theta \in \mathbb{R}^n$. The ray can be parameterized as:

$$R = \{\mathbf{y}_0 + t\mathbf{v}_\theta : t \geq 0\}, \tag{36}$$

where $t \geq 0$ is the parameter along the ray. Since $\mathbf{y}_0 \in \text{int}(C)$, the ray $R$ starts inside $C$. By the definition of convex, the ray $R$ must intersect $C$, and the portion of the ray close to $\mathbf{y}_0$ is entirely contained in $C$. Let $t^*$ be the supremum of the set of parameters $t \geq 0$ such that $\mathbf{y}_0 + tv \in \text{int}(C)$ which is given as:

$$t^* = \sup\{t \geq 0 : \mathbf{y}_0 + tv \in \text{int}(C)\}. \tag{37}$$

Since $\text{int}(C)$ is open and $\mathbf{y}_0 \in \text{int}(C)$, this set is non-empty and $t^*$ exists. Define:

$$\mathbf{y}_1 = \mathbf{y}_0 + t^*v. \tag{38}$$

By construction, the point $y_1$ satisfies the following: (1) For any $t < t^*$, $\mathbf{y}_0 + t\mathbf{v} \in \text{int}(C)$; (2) For any $t > t^*$, $\mathbf{y}_0 + t\mathbf{v} \notin C$. Therefore, $\mathbf{y}_1$ lies on the boundary of $C$, i.e., $\mathbf{y}_1 \in \partial C$. Furthermore, since the ray is continuous and $C$ is closed (as the finite intersection of closed sets), $\mathbf{y}_1 \in C$. For any ray originating from $\mathbf{y}_0$, the closest intersection point $\mathbf{y}_1$ belongs to $C$ and lies on its boundary $\partial C$.

□

## G   TOWARD THE NON-CONVEX CASE

When the constraint set is non-convex, the boundary induces a discontinuous, non-homeomorphic angle map. To apply HoP in this setting we reformulate (7) by piecewise-linear homeomorphic extension (Rourke & Sanderson, 2012) Eq. equation 39, which skips the infeasible region. To illustrate HoP in non-convex case, Fig. 5 provides a visualized algorithm for (39). Let $\mathbf{s}_t$ be the $t$-th intersection of the ray starting at $\mathbf{y}_0$ in the direction $\mathbf{v}_\theta$ with the boundary, ordered by increasing distance ($t = 1, 2, \ldots, N$). Additionally, each intersection point $\mathbf{s}_t$ corresponds to an angle $\phi_t = \arctan \|\mathbf{s}_t - \mathbf{y}_0\|_2$, defined as the angle between the vector from the origin to $\mathbf{s}_t$ and the z-axis in Fig. 2 (b). As shown in Fig. 5, each effective interval, $[\mathbf{s}_{2\tau}, \mathbf{s}_{2\tau+1}]$, and the corresponding effective angle width, $[\phi_{2\tau}, \phi_{2\tau+1}]$, lies inside the feasible region, where $\tau \in 0, 1, \cdots, (N-1)/2$. By contrast, the following interval $[\mathbf{s}_{2\tau+1}, \mathbf{s}_{2(\tau+1)}]$ is infeasible. This alternation follows from the Jordan-curve theorem and the parity rule (Hales, 2007), which indicates $N$ must be odd.

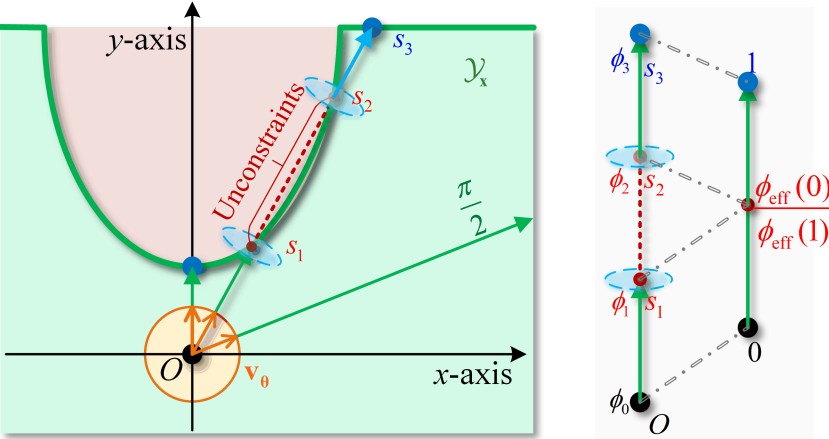

Figure 5: Principal of HoP in non-convex constraints problem

$$\psi = \varphi_{\text{eff}}(\frac{N-1}{2})\bar{z}_r - \varphi_{\text{eff}}(\tau - 1) + \phi_{2\tau}, \quad \text{if } \frac{\varphi_{\text{eff}}(\tau - 1)}{\varphi_{\text{eff}}(\frac{N-1}{2})} \leq \bar{z}_r < \frac{\varphi_{\text{eff}}(\tau)}{\varphi_{\text{eff}}(\frac{N-1}{2})}, \qquad (39)$$

$$\varphi_{\text{eff}}(\tau) = \sum_{n=0}^{\tau} \phi_{2n+1} - \phi_{2n}, \qquad (40)$$

Eq. equation 39 defines a homeomorphic map from the scaled variable $\bar{z}_r$ to the angle $\psi$ by proportional scaling, while $\varphi_{\text{eff}}(\tau)$ accumulates the effective angular widths, $\phi_{2n+1} - \phi_{2n}$ where $n \in 0, 1, \cdots, \tau$. The detailed derivation of piecewise-linear mapping in 39 is given in Appendix G.1. Additional engineering implementation, including handling redundant constraints and improving computational efficiency, are provided in Appendix E.

### G.1   DERIVATION OF THE PIECEWISE–LINEAR GLUING EXTENSION

Let $\mathbf{y}_0$ denote the polar origin and $\mathbf{v}_\theta$ a unit-direction vector. Consider the half-line

$$\{\mathbf{y}_0 + t\,\mathbf{v}_\theta : t \geq 0\}$$

intersecting the (possibly non-convex) boundary $\partial \mathcal{Y}$ at exactly $N$ points $\mathbf{s}_1, \ldots, \mathbf{s}_N$, ordered by increasing Euclidean distance:

$$\|\mathbf{s}_1 - \mathbf{y}_0\|_2 < \|\mathbf{s}_2 - \mathbf{y}_0\|_2 < \cdots < \|\mathbf{s}_N - \mathbf{y}_0\|_2.$$

Define the associated "radial angles"

$$\phi_t = \arctan\|\mathbf{s}_t - \mathbf{y}_0\|_2, \quad t = 0, 1, \ldots, N,$$

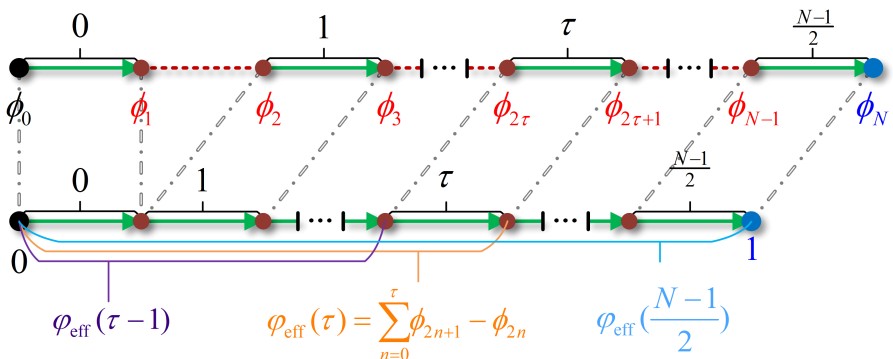

Figure 6: Piecewise-linear homeomorphic extension. Upper row: raw intersections $s_t$ and angles $\phi_t$. Lower row: feasible segments (green arrows) and cumulative effective angles $\varphi_{\text{eff}}(\tau)$.

with $\phi_0 = 0$. In the semi-unbounded case we adopt the convention that "infinity" constitutes the $N$th intersection, $\phi_N = \lim_{r\to\infty} \arctan(r) = \frac{\pi}{2}$.

As illustrated in Figure 6, only the intervals between even and odd intersections—$s_{2\tau} \to s_{2\tau+1}$—lie inside the feasible set. The $\tau$th feasible angular width is

$$\Delta\phi_\tau = \phi_{2\tau+1} - \phi_{2\tau}, \qquad \tau = 0, 1, \ldots, \frac{N-1}{2},$$

and the cumulative effective angle up to index $\tau$ is precisely Eq. equation 40,

$$\varphi_{\text{eff}}(\tau) = \sum_{n=0}^{\tau} \big(\phi_{2n+1} - \phi_{2n}\big).$$

Hence the global effective range is $\varphi_{\text{eff}}\big(\frac{N-1}{2}\big)$.

We now construct a strictly increasing map $\bar{z}_r \in [0,1] \mapsto \psi \in [0, \varphi_{\text{eff}}((N-1)/2)]$. Define breakpoints

$$T_\tau = \frac{\varphi_{\text{eff}}(\tau)}{\varphi_{\text{eff}}\big(\frac{N-1}{2}\big)}, \quad \tau = -1, 0, \ldots, \frac{N-1}{2},$$

with $T_{-1} = 0$ and $T_{(N-1)/2} = 1$. For any $\bar{z}_r \in [T_{\tau-1}, T_\tau)$, we linearly interpolate

$$\bar{z}_r = T_{\tau-1} \mapsto \varphi_{\text{eff}}(\tau - 1), \quad \bar{z}_r = T_\tau \mapsto \varphi_{\text{eff}}(\tau),$$

yielding

$$\psi = \varphi_{\text{eff}}(\tau - 1) + \frac{\bar{z}_r - T_{\tau-1}}{T_\tau - T_{\tau-1}} \big[\varphi_{\text{eff}}(\tau) - \varphi_{\text{eff}}(\tau - 1)\big].$$

Upon substituting $T_\tau = \varphi_{\text{eff}}(\tau)/\varphi_{\text{eff}}\big(\frac{N-1}{2}\big)$ and algebraic rearrangement we recover Eq. equation 39:

$$\psi = \varphi_{\text{eff}}\big(\tfrac{N-1}{2}\big)\bar{z}_r - \varphi_{\text{eff}}(\tau - 1) + \phi_{2\tau},$$

valid whenever $\frac{\varphi_{\text{eff}}(\tau-1)}{\varphi_{\text{eff}}(\frac{N-1}{2})} \leq \bar{z}_r < \frac{\varphi_{\text{eff}}(\tau)}{\varphi_{\text{eff}}(\frac{N-1}{2})}$.

Crucially, by the parity rule of the Jordan curve theorem (Hales, 2007), any ray from the interior of a simple closed curve intersects it an odd number of times, ensuring $N$ is odd and that feasible/infeasible

segments alternate. Treating infinity as the final intersection seamlessly extends the construction to semi-unbounded constraint sets without altering the homeomorphic map of Eq. equation 39.

## G.2 APPLICATION: NON-CONVEX PARABOLIC CONSTRAINTS

Table 2: Experimental results over Non-convex parabolic problem

| Problem | Methods | Obj. Value ↓ | Max. Cons ↓ | Mean. Cons ↓ | Vio. Rate ↓ | Time / ms ↓ |
|---|---|---|---|---|---|---|
| **Non-convex parabolic constrained problem** | Optimizer | $-0.1817$ | 0.0000 | 0.0000 | 0.00% | 0.99292 |
| | HoP (+R) | **$-0.2004$** | 0.0000 | 0.0000 | 0.00% | 0.03058 |
| | HoP (−R) | $-0.0000$ | 0.0000 | 0.0000 | 0.00% | 0.03065 |
| | NN-SSL | $\underline{-0.1977}$ | $\underline{0.0555}$ | $\underline{0.0155}$ | $\underline{55.53\%}$ | $\underline{0.00692}$ |
| | NN-SL | $-0.0050$ | 0.0000 | 0.0000 | 0.00% | 0.00691 |
| | NN-SSL-SC | $\underline{-0.1977}$ | $\underline{0.0549}$ | $\underline{0.0152}$ | $\underline{55.03\%}$ | $\underline{0.00701}$ |
| | NN-SL-SC | $-0.0012$ | 0.0000 | 0.0000 | 0.00% | **0.00692** |
| | DC3 | $-0.1900$ | 0.0000 | 0.0000 | 0.00% | 0.03083 |
| | HomeoProj | – | – | – | – | – |

To demonstrate HoP's effectiveness on complex non-convex problems, we consider a sinusoidal QP objective from Eq. equation 14 subject to two opposing paraboloidal constraints:

$$\mathbf{y}^T \mathbf{A}_i' \mathbf{y} + \mathbf{b}_i^T \mathbf{y} + c_i \leq 0, \quad i = 1, 2 \tag{41}$$

where $\mathbf{A}_i' \in \mathbb{R}^{(n-1) \times (n-1)}$ are positive-definite matrices determining curvature in the first $n-1$ coordinates. The first constraint ($i = 1$) opens downward along the $y_n$-axis providing a lower bound, while the second opens upward providing an upper bound. Their intersection defines a non-convex, semi-unbounded feasible region extending the structure in Liang et al. (2023) to high dimensions.

As shown in Table 2, HoP achieves the best objective value of $-0.2004$ with 0% constraint violation, outperforming both the optimizer ($-0.1817$) and DC3 ($-0.1900$). Neural network baselines slightly improve objectives but exhibit 55% violations. Moreover, HOP runs over 30× faster than traditional optimizers, demonstrating the practical efficiency of our piecewise-linear extension for challenging non-convex problems.

