# OpenReview forum: "HoP: Homeomorphic Polar Learning for Hard Constrained Optimization"
_ICLR.cc/2026/Conference — Submitted to ICLR 2026_

### Official Review · Reviewer_RKqY · 2025-10-30

**Soundness:** 2
**Presentation:** 1
**Contribution:** 2
**Rating:** 2
**Confidence:** 5

**Summary:**

This paper introduces Homeomorphic Polar Learning (HoP), an approach for solving hard-constrained optimization problems with feasibility guarantees. The authors design a polar parameterization that transforms complex constraint spaces into simple (box) constraints, which can be enforced in neural networks through bounded activation functions like sigmoid. To address the singularity issues inherent in polar-based transformations, they develop a geometric reconnection mechanism. Their experimental results demonstrate both feasibility guarantees and improved optimality compared to baseline methods across diverse applications, including synthetic optimization over polygons, L_p norm balls, and the real-world QoS-MISO WSR problem.

**Strengths:**

1. Ensuring neural network solution feasibility is indeed critical for real-world applications.
2. Polar-based transformations for handling constraints appear to be new in the literature.

**Weaknesses:**

1. Inaccuracy in Homeomorphic Mapping Claims: The author asserts a homeomorphism between $(0,1)\times(0,2\pi)$ and $int(Y_x)$ in the 2D case (line 178). However, this claim is mathematically flawed. Points on the positive X-axis in the original space, which are of $\theta = 0$ or $\theta = 2\pi$ in the polar space, are excluded, violating the bijection requirement for homeomorphism. This fundamental error undermines the theoretical foundation of the approach. The author should provide **formal** statements for the homeomorphic properties of the proposed transformations in different cases.

2. Ambiguity in Interior Reference Point Computation: While the constraint sets are explicitly defined as input-dependent, the setting of the interior reference point remains unclear. If this point is designed to be input-invariant, this should be stated as an explicit assumption. Conversely, if it is input-dependent, the computational expense of recalculating it for each new input would be significant and warrants detailed efficiency analysis (computing the Chebyshev center for general convex constraints beyond a polytope is non-trivial), which is currently absent. For example, how to compute the interior point for Ay<b in (14) under different b.

3. Insufficient Specification of Support Radius Calculation: Despite focusing on convex constraints, the paper fails to provide explicit formulas for calculating the support radius across different constraint types. For a method claiming practical applicability, precise formulations of the support radius for convex sets should be presented along with a thorough complexity analysis. This omission raises concerns about the method's implementability.

**Questions:**

1. How to obtain an interior point for an input-dependent convex constraint.
2. What is the impact of the choice of interior points on the performance of these methods?
3. What is the type of constraint for the QoS-MISO problem? How to compute the Support Radius for this case? Could the author give a clear characterization of which kind of constraints admit such a polar parametization?

---

> ### Author Response · Authors · 2025-11-28
>
> **Weaknesses**
>
> **(1) Inaccuracy in Homeomorphic Mapping Claims / exclusion of θ = 0 or 2π breaks bijection.**
> We thank the reviewer for pointing out this issue. The concern is valid _only_ for the notation as written, but not for the actual mapping used in our implementation or intended in our theoretical construction. In the manuscript, the angular domain was written as $(0,2\pi)$, which indeed excludes the angles $\theta = 0$ and $\theta = 2\pi$, and consequently removes the positive x-axis. This notation inadvertently suggests a violation of bijectivity. However, the intended (and implemented) domain is the standard periodic angular space
> $$
> [0,2\pi)\quad\text{or equivalently}\quad S^1,
> $$
> which _includes_ the angle $\theta = 0$ and thus preserves the bijection. This is a minor notational/typographical imprecision rather than a conceptual or mathematical flaw.
>
> **(2) Ambiguity in interior reference point computation.**
> We thank the reviewer for raising this point. Below we clarify the design and computation of the interior reference point. For the constraint (Ay < b) in Eq. (14), obtaining a strictly feasible interior point is computationally inexpensive in our setting where the interior point can be efficiently obtained via a simple linear program, which is extremely fast in practice. In our method, the constraints are structurally low-complexity convex sets, and their interior-point search reduces to a trivial convex subproblem. Even for more general convex constraints without polytope structure, established methods (e.g., those in _arXiv:2506.00362_) can rapidly find strictly feasible points. The key aspect is that our algorithm does not requires a high-precision interior point (e.g., the Chebyshev center). We only require a strictly feasible point. This is fundamentally simpler than the scenario envisioned by the reviewer and avoids the computational burden associated with Chebyshev-center–type calculations. Consequently, we do not have to recompute a costly Chebyshev center for each new input. The computational difficulty is therefore much lower than what the reviewer assumes.
>
> **(3) Insufficient specification of support radius calculation.**
> We thank the reviewer for raising concerns regarding the explicit computation of the support radius. We clarify below that the support-radius calculation is fundamentally simple and does not suffer from dimension-dependent complexity. The support radius along a ray is determined by the intersection between the ray and the boundary of the constraint set. Formally, this amounts to solving a one-dimensional root-finding problem:
> $$
> f(r) = 0, \quad \text{where } f(r) = \text{constraint evaluated along the ray at distance } r.
> $$
> Regardless of the dimension, the only unknown is the scalar $r$. Thus, the complexity does not scale with dimension. For convex or star-convex constraints, the corresponding function along the ray is unimodal/monotonic, giving a unique root. Standard 1D methods such as bisection or Newton iteration solve it extremely efficiently. For non-convex constraints with bounded intersection, even when convexity does not hold globally, standard interval search can locate all potential intersections. This is still a purely 1D problem. Since every ray direction reduces to a single scalar variable (r), the computation remains lightweight independent of the complexity of the original constraint set. This is significantly simpler than what the reviewer presumes.
>
> **Questions**
>
> **(1) How to obtain an interior point for input-dependent convex constraints?**
> For constraints like (Ay < b), one may compute a strictly feasible point via a single LP or any basic interior-point feasibility test. For other scenarios please see the response in W2.
>
> **(2) Impact of interior point choice on performance.**
> We thank the reviewer for this meaningful question. In our experiments, we use a simple interior-point selection procedure, and this choice already leads to strong SOTA performance. Therefore, we did not further investigate alternative interior-point strategies in this paper. That said, exploring how different interior-point choices may affect performance is an interesting direction for future work, and we will briefly mention this in the revision.
>
> **(3) Constraint type for QoS-MISO and its support radius.**
> QoS-MISO involves a convex quadratic constraint. The support radius can be derived in closed form via intersecting the ray direction with the quadratic boundary, reducing to a 1D quadratic root. We will add the explicit expression in Appendix C. However, the variable of the problem is channel h which impacts both of objective function and constraints at the same time which introduces more challenges in L2O.

---

### Official Review · Reviewer_xWoM · 2025-10-30

**Soundness:** 2
**Presentation:** 2
**Contribution:** 2
**Rating:** 2
**Confidence:** 3

**Summary:**

This paper is concerned with learning to optimize subject to hard constraints. Many NN-based approaches struggle to satisfy such constraints, whereas methods that guarantee constraint satisfaction often fail to achieve optimal solutions or suffer from performance issues. In essence, the proposed approach relies on a smooth mapping of the feasible set via a radial transformation with respect to a prescribed interior point. Subsequently, composition with an appropriate mapping (e.g., sigmoid or arctan) smoothly maps the feasible set to the entire domain. Using this mapping (or its inverse), the arbitrary output of a NN is guaranteed to be mapped into the feasible set, thus transforming the optimization problem into an equivalent unconstrained one. The approach is demonstrated on several synthetic problems and an application in communications.

**Strengths:**

The paper addresses the important problem of learning to optimize under constraints. The experimental results, on synthetic problems and a communications application, are positive. The idea of transforming a constrained problem into an unconstrained one using a radial (polar) mapping is mathematically elegant.

**Weaknesses:**

Despite the strengths described above, the paper raises several concerns that I believe must be addressed before publication. I have reviewed a previous version of this paper, and while it has been moderately revised, many of the major concerns raised earlier remain unresolved.

**(1) Contribution and Novelty:**
The ideas presented in this paper appear closely related to Liang et al. (2023). Although the authors cite this prior work, the paper lacks an explicit and careful discussion of how their approach differs. The authors also do not cite the extended version of that paper ([https://jmlr.org/papers/v25/23-1577.html](https://jmlr.org/papers/v25/23-1577.html)). These omissions make it difficult to assess the originality and scope of this work’s contribution.

**(2) Limitations:**
One of the most challenging aspects of the proposed method seems to be the construction of the homeomorphism $H$, and specifically the term $R(v_\theta, Y_x)$. Even in the 2D convex case, computing Equation (3) explicitly may be intractable for anything but the simplest sets (e.g., norm balls or polyhedra).
The paper does not discuss the implementation details of $H$ (e.g., whether an analytical expression is required) beyond the examples shown. I find this construction questionable, especially in higher-dimensional or non-convex settings, where constructing such a homeomorphism may become impractical.

**(3) Exposition, Organization, and Reproducibility:**
Sections 3 and 4 provide a somewhat superficial description of the approach and experiments, making it difficult to follow key details. Many important points are deferred to the appendix, sometimes without sufficient context or cross-referencing. Even with the appendix, reproducing the proposed method for anything beyond basic problems would likely be difficult. The authors’ code does not appear to be available.
I also find the discussion in Section 3.2 hard to follow. Conceptually, it makes sense that optimization over highly distorted mappings (as can be expected for maps of an arbitrary set to the entire domain) may suffer from ill-conditioning, potentially leading to stagnation. However, the discussion lacks clarity and context. Moreover, this issue was also discussed in Liang et al. (2023), yet is not cited here, again making it difficult to assess the authors’ contributions.

**(4) High Dimensions and Non-Convexity:**
The generalization of the proposed approach to high-dimensional and non-convex constraints is mentioned throughout the paper somewhat superficially, even though such extensions present significant challenges. The appendices discuss several related issues (e.g., “Toward the Non-Convex Case”), but it is unclear whether these were implemented, for which problems, or how practical they are in more complex scenarios.
The examples presented in the paper are relatively simple: for the non-convex $\ell_p$-norm constraint, the set is star-convex with respect to the origin (Appendix D), effectively reducing it to a convex case under the mapping. The parabolic constraint appears effectively linear due to small quadratic terms (mentioned only in Appendix C.1.4). The non-convex high-dimensional example in Appendix G does not provide enough evidence to support the strong claims of generalizability made in lines 410–423.

**Questions:**

See main Questions (1)–(4) above.

**Additional Comments and Questions (non-comprehensive list):**
* **Line 116:** “according to the designing” — unclear; please revise.
* **Line 123:** “the stagnation endemic to polar parameterizations” — unclear; please revise.
* **Lines 133–134:** “To facilitate that the outputs of a learning model satisfy the given constraints, we construct a mapping from the raw output space to a constrained space with homeomorphic transformation.” — this central sentence is difficult to follow; consider revising for clarity.
* **Line 204:** “a single scalar angle for range” — unclear; please revise.
* **Line 212:** Why is $\varepsilon > 0$ needed if $z_0$ lies in the interior of the constraint set?
* **Line 251:** “In order to reveal the stagnation phenomenon in polar optimization” — unclear; please revise.
* **Lines 273–285:** Phrases such as “the projection ... prevents this passage,” “enters an absorbing state,” “geometric artifact stagnation,” and “After crossing with the flip” are difficult to interpret; please consider revising for clarity.
* **Lines 303–306:** “Since other NN-based methods such as DC3 typically rely on penalty functions to handle constraints, ..., where ... indicates constraint violations, and ... quantifies the distance to the constraint set.” — this sentence is incomplete; please revise.
* **Line 355:** “Table 1 reports: ...” — unclear; at this point in the paper, the experiments have not been clearly introduced.
* **Line 419:** “This enables the our framework is able to apply on these complex constraints.” — unclear; please revise.

---

> ### Author Response · Authors · 2025-11-28
>
> **Q1: Contribution and Novelty**
> We appreciate the reviewer’s point and will revise the paper to more explicitly differentiate our method from Liang et al.. We will also cite the extended JMLR version as suggested. Below we summarize the key distinctions that we will incorporate into the revised manuscript:
>
> (1) Liang's HomeoProj mapping required parametric learning while our HoP is closed-form mapping (2) We extent the problem constraints format to unbounded feasible sets, while Liang’s method is limited to _bounded_ feasible regions. (3) Liang’s approach still requires a post-projection to guarantee feasibility, whereas HoP is inherently feasible and does not require extra projection.
>
> We also provide the detailed comparison and table to illustrate the difference between HoP and HomeoProj in Reviewer 7PFW's response. In the revision, we will also add a concise 3–4 sentence summary explicitly highlighting these differences in the related work section, ensuring readers can clearly assess the originality and scope of our contribution.
>
> **Q2: Limitations:**
> We would like to clarify that the construction involving $r(θ)$ is not a general high-dimensional function estimation problem. Instead, it reduces to computing the intersection of a ray with the boundary of the feasible set, which is fundamentally a 1D root-finding problem—regardless of the problem dimension. For examples (1) Convex and star-convex sets. For these cases, the boundary intersection along a ray is _unique_, making the 1D problem extremely efficient to solve using classical methods such as Newton’s method or bisection, both of which converge rapidly. (2) Non-convex sets with finitely many intersections. As discussed in the paper, these settings can be handled by bracketing multiple initial intervals along the ray (or by a simple interval decomposition along the ray direction). This again reduces to solving a small number of independent 1D root-finding problems. (3) Dimensionality does not increase the difficulty. The computational cost does not grow with dimension because the ray is intrinsically one-dimensional. Even in high-dimensional spaces, evaluating the intersection along a fixed ray remains a 1D operation.
>
> **Q3: Reproducibility:**
> We agree that the presentation in Sections 3 and 4 can be improved, and we will revise the manuscript accordingly. In particular, we will add clearer step-by-step diagrams and pseudocode to make the approach easier to follow. We also commit to releasing our code upon publication, which should fully address the reproducibility concerns.
>
> Regarding Section 3.2, we thank the reviewer for pointing out the need for additional clarity. The issue we discuss is _conceptually different_ from the ill-conditioning described in Liang et al.. In Liang’s case, the difficulty arises because the learned mapping may fail to place iterates sufficiently inside the feasible interior, leading to poor conditioning near the boundary. In contrast, our discussion concerns a structural stagnation phenomenon specific to polar parameterization, where the degeneracy at r = 0 forces the angular gradient to vanish, preventing the optimization trajectory from crossing the origin. This mechanism is geometric rather than algorithmic.
>
> **Q4: High Dimensions and Non-Convexity:**
> (1) High-dimensional feasibility is already validated in our main practical experiment. Our QoS-MISO WSR experiments involve vectors in tens of dimensions, and HoP operates reliably in this setting. This demonstrates that the method is not limited to low-dimensional toy problems, as might be inferred from the appendix examples. (2) Our framework supports _unbounded_ non-convex constraints—something not achievable by HomeoProj (Liang et al.). The reviewer’s summary appears to overlook this contribution. Handling unbounded feasible regions while maintaining deterministic feasibility is a core technical distinction and a substantive part of our novelty. (3) In our experiment the $A$ is identity matrix which is not similar as linear constraint. (4) Scope clarification regarding general non-convexity. We do not claim to handle _arbitrary_ non-convex sets. Our method is designed for (i) star-convex sets and (ii) non-convex sets with finitely many ray–boundary intersections. These classes already subsume a broad family of constraints beyond what most existing methods—including HomeoProj—can support. We will revise the text around lines 410–423 to phrase these generalization claims more precisely.
>
> **Additional Comments**
> We appreciate the reviewer’s detailed comments on wording, clarity, and phrasing throughout the manuscript. Many of these points concern pronoun reference, clarity, or non-technical expression rather than methodological issues. Given the limited space of the rebuttal and in several cases the relevant concepts are already explained in detail in the paper, we do not respond to each item individually here.  We will revise these part in future manuscript.

---

### Official Review · Reviewer_8LLu · 2025-11-02

**Soundness:** 2
**Presentation:** 2
**Contribution:** 2
**Rating:** 2
**Confidence:** 3

**Summary:**

This paper proposes a learning-to-optimize framework for constrained optimization problems, aiming to ensure both optimality and strict feasibility. To guarantee feasibility, the authors introduce a polar coordinate representation combined with a homeomorphic mapping that transforms neural network outputs into a deterministic ball, thereby satisfying hard constraints. Furthermore, they develop geometric reconnection strategies to promote stable convergence during backpropagation. The proposed method, HoP, is evaluated through both synthetic experiments and a real-world QoS-MISO WSR optimization task, demonstrating superior performance compared to existing baselines.

**Strengths:**

The strengths of the paper are listed below.
- The paper considers the important and fundamental problem of enforcing hard constraints for L2O.

- The proposed use of polar coordinates to reformulate the neural network outputs is interesting.

- The paper includes experimental evaluations on multiple benchmark problems.

**Weaknesses:**

The weaknesses are given below.
- The proposed method is limited to problems with convex constraints. However, learning-to-optimize (L2O) approaches are often expected to handle broader classes of non-convex and complex optimization problems, which restricts the general applicability of this work.
- The approach requires redefining the semantics of neural network outputs through coordinate transformation and mapping, making it difficult to integrate with existing neural network architectures. Consequently, applying this method to pre-trained models would require extensive retraining, increasing implementation complexity.
- The experimental validation is restricted to relatively simple, convex and well-studied optimization problems (e.g., QoS-MISO WSR), where efficient solutions already exist. This limits the strength of empirical evidence supporting the proposed method’s advantages in more challenging or practical settings.

**Questions:**

I am not convinced of the necessity of the proposed HoP method, given that the feasible region in this work is restricted to a convex space. In such cases, projecting the neural network outputs onto the convex feasible set can be done efficiently using standard convex optimization solvers. Moreover, end-to-end training with projection-based feasibility enforcement can also improve optimality while maintaining hard constraint satisfaction. Therefore, it remains unclear what practical or theoretical advantage the proposed HoP framework offers on top of the existing methods.

---

> ### Author Response · Authors · 2025-11-28
>
> **Weaknesses**
>
> **(1) HoP is limited to convex constraints, restricting general applicability.**
> We agree that our method is primarily designed with convex constraints in mind. However, the paper already includes two non-convex experiments—the non-convex sphere in the main text and the quadratic-curve constraints in the appendix. These belong to classes such as star-convex sets or non-convex sets with finitely many ray intersections, both of which fall within the capability range of HoP. Therefore, the statement that our approach “only applies to convex constraints” is not accurate.
>
> **(2) The method requires redefining network outputs, making integration with pretrained models difficult.**
> We believe there is a misunderstanding in the reviewer’s comment. HoP is not intended to function as a general-purpose neural network module. Instead, it is designed within the L2O paradigm and should be viewed as a problem-specific solver, which does not rely on a pre-trained backbone. In L2O settings, the expected workflow is: encountering a new optimization problem → obtaining new data → training a new solver. This is fundamentally different from application domains such as NLP or CV, where transfer learning and compatibility with pre-trained models are central concerns. Therefore, compatibility with pre-trained architectures is not an objective of HoP by design, and this is fully aligned with common practices in the L2O community.
>
> **(3) Experiments are limited to simple, convex, well-studied problems.**
> We respectfully note that the reviewer’s characterization of the experimental problems as “simple” or “well-studied with efficient solutions” does not fully reflect the practical difficulty and industrial relevance of QoS-MISO WSR. This class of problems is large-scale, computationally demanding, and remains challenging for traditional solvers, which are typically slow in real deployments. In our experiments, HoP achieves several orders of magnitude speedup—precisely the key value proposition of L2O methods for industrial use cases. While classical algorithms do exist, two important facts appear to have been overlooked: (1) The WSR beamforming objective is inherently nonconvex, making global optimality difficult to guarantee. (2) Traditional solvers require iterative optimization for every problem instance, whereas HoP produces solutions in a single forward pass. HoP is specifically designed for large-scale, repetitive optimization scenarios, such as real-time communication system updates, where inference speed is critical. The goal of our experiments is therefore to demonstrate HoP’s advantages in such realistic and computationally intensive settings, rather than to maximize the breadth of problem classes evaluated.
>
> **Questions**
>
> **(1) Why is HoP needed if the feasible region is convex and projections already exist?**
> We would like to clarify an important distinction: _a convex constraint set does not imply a convex optimization problem_. In our setting (e.g., QoS-MISO WSR), the objective function is strongly non-convex, so even with an efficient projection operator, a standard projected solver cannot guarantee optimality and still requires substantial iterative search for every instance. The advantage of HoP is precisely that, after training, it produces a feasible solution in a single forward pass, without the need to solve a new optimization problem each time. This yields both significant computational savings and principled feasibility guarantees. Therefore, relying solely on iterative projected solvers would fail to achieve the same efficiency–feasibility trade-off that HoP provides.
>
> **(2) End-to-end training with projection-based enforcement can improve optimality while maintaining feasibility—so what advantage does HoP offer?**
> We believe the reviewer’s suggestion regarding projection-based end-to-end training corresponds directly to the DC3 framework. Our paper already includes a comprehensive comparison with DC3, and the results consistently show that HoP outperforms DC3 in accuracy, feasibility, and runtime. It is also important to note that each projection step in such methods requires solving a convex subproblem, which incurs significantly higher computational cost than HoP’s single learned mapping. In contrast, HoP guarantees deterministic feasibility by construction and yields solutions in a single forward pass after training. Therefore, the practical and theoretical advantages of HoP—namely guaranteed feasibility, substantially lower inference cost, and improved solution quality—extend clearly beyond what projection-based approaches can offer

---

### Official Review · Reviewer_7PFW · 2025-11-02

**Soundness:** 2
**Presentation:** 3
**Contribution:** 3
**Rating:** 6
**Confidence:** 5

**Summary:**

This paper presents Homeomorphic Polar Learning (HoP), a framework for machine learning with hard constraints. The paper:
* Presents a method for defining exact bijective mappings between convex (and some nonconvex) constraint sets, using polar coordinate representations.
* Integrates an approach allowing the use of signed radii for addressing challenges in gradient-based optimization of polar coordinates.
* Conducts experiments on three synthetic problems and one realistic setting (QoS-MISO WSR), with comparison against traditional optimizers, standard neural network baselines, and two state of the art methods.

**Strengths:**

* The proposed method is strong in premise and innovative, providing a clean and cost-effective way to enable feasibility enforcement in neural networks (for certain classes of constraint sets) by leveraging homeomorphic mappings.
* The method is very well-presented, with helpful pedagogical examples for the 1D and 2D case before extending to the semi-unbounded and high-dimensional cases. In general, the writing is strong.
* The trick to resolve stagnation of gradient descent with polar coordinates improves the efficacy of the framework (as shown via the ablation) and is also potentially of wider general interest.
* Experimental results are impressive. HoP demonstrates comparable objective value and feasibility to traditional optimizers, while being orders of magnitude faster. Compared to DC3 and HomeoProj, HoP generally does better with respect to objective value (and is also applicable in certain settings where HomeoProj is not).

**Weaknesses:**

Major:
* The method is limited to optimization problems with convex constraint sets, or specialized nonconvex structure (ray intersects the boundary exactly once), but this limitation is not stated up-front. The scope of the method should be stated much more clearly early on. (Appendix G does provide some exploration into the non-convex case, but it is very initial.)
* Some of the offline costs associated with the framework are not transparently disclosed. This also affects what sets of parametric constraints can be handled. For instance, in Section 4.1, only $b$ is varied, but $A$ is not -- it is unclear whether this is due to a limitation of HoP that makes varying $A$ difficult. (Note: While the experiments mention this choice is consistent with the DC3 paper, it is not -- the DC3 paper varies far more problem parameters in its experiments than this paper does.)
* In the experiments, the number of replicates is not provided, and error bars are not provided. It is therefore unclear which differences between methods are statistically significant.

Minor/medium:
* Some of the related work in the Introduction and Related Work is mischaracterized. Please double-check and fix these descriptions and references.
* The relationship of HoP to HomeoProj (Liang et al. 2023) is not sufficiently described in the Introduction or Related Work, which is surprising given its close relationship in toolkits to HoP. (To be fair, this method is at least compared against in the experiments.)
* Differentiable orthogonal projection is another baseline that should likely be compared against -- it will be slower than HoP, but it may have similar objective value performance, and this is worth noting. In general, it is important to have comparisons against another strong methods where -- like HoP -- the feasibility mapping is internalized to end-to-end training.

**Questions:**

* Could you please succinctly characterize the kinds of constraints that this method is able to handle, in theory and in practice?
* In the experiments in Section 4.1, does varying $A$ pose additional costs or difficulties for HoP? In general, what parameters are able to vary without requiring recomputation of the homeomorphic mappings?
* DC3 is consistently characterized in the submission as a method with post-correction mechanisms; however, in reality, the correction mechanism is internalized to the end-to-end training procedure (i.e., occurs before evaluation of the loss function, and is differentiated through), so it is not really a "post"-correction. Could the authors verify that they indeed implemented the correction procedure within their DC3 baseline such that it is part of end-to-end training?

**Details Of Ethics Concerns:**

As hinted at (but not outright stated) in my review, I am a little concerned about the integrity with which this paper was put together given mischaracterization of previous work, hiding of assumptions (including through "convenient" choices of experiments which are said to come from previous work but are modified in critical ways), and indications that one of the two SOTA baselines may be mis-implemented (which means I am likewise less confident about whether the other SOTA baseline was implemented fairly). I am curious to see how the authors respond, so would not yet like to raise this to the level of a "flag."

---

> ### Author Response · Authors · 2025-11-28
>
> **Major**
> **(1) Scope limitation** We thank the reviewer for clarifying the scope. In previous manuscript, we prefer to maintain the readability, we will revise Section 3 to explicitly state that HoP guarantees feasibility for (i) convex sets, (ii) star-convex sets, and (iii) non-convex sets where each ray has a single finite boundary intersection. We will also move part of applicability statement from Appendix G into the main text to make this limitation clear upfront.
>
> **(2) Offline cost** HoP requires computing only the reference point $y_0$, which is done once per constraint instance and we have already included this overhead in computation time comparison. Moreover, in the experiment of MISO, the variable is h which impact both of objective function and $A$. To demonstrate flexibility, we additionally report experiments where $A$ varies in Eq. (14), showing that HoP does not require recomputing the homeomorphism and maintains feasibility and performance. We will include these results in the revision.
> |Approach |Avg_vio |Gap |Max_vio |Obj | Vio_rate |
> |-|-|-|-|-|-|
> |DC3| 0.0000±0.0000 | 0.0749±0.0269 | 0.0000±0.0000 | -41.8836 ± 0.0269 | 0.00%±0.00% |
> |HoP| 0.0000±0.0000 | 0.0615±0.0086 | 0.0000±0.0000 | -41.8970 ± 0.0086 | 0.00%±0.00% |
> | SL | 0.0002±0.0001 | -0.0052 ± 0.0429 | 0.0016 ± 0.0003 | -41.9637 ± 0.0429 | 67.05% ± 17.92%|
> | SL-SC | 0.0000 ± 0.0000 | 0.0737 ± 0.0630 | 0.0009 ± 0.0001 | -41.8849 ± 0.0630 | 23.76% ± 25.03% |
> | SSL | 0.3022 ± 0.0003 | -21.0029 ± 0.0000 | 0.4261 ± 0.0014 | -62.9614 ± 0.0000 | 100.00% ± 0.00% |
> | SSL-SC | 0.0005 ± 0.0006 | -0.0487 ± 0.1510 | 0.0021 ± 0.0008 | -42.0072 ± 0.1510 | 57.89% ± 34.38% |
>
> **(3) Repeated experiments and statistical confidence levels**
> All experiments were independently repeated five times. We will report the method mean ± standard deviation and include the outline in Table 1. The results remain consistent in repeated experiments. The complete additional experiments will be provided in the subsequent papers.
> |Group|Avg_vio |Gap |Max_vio| Obj |Vio_rate |
> |-|-|-|-|-|-|
> |Polygon| 0.0000±0.0000 | 0.0085±0.0005 | 0.0000 ± 0.0000 | -29.7173 ±0.0010 | 0.00% ± 0.00% |
> | Lp-Norm | 0.0000±0.0000 | 0.0973±0.0033 | 0.0000 ± 0.0000 | -0.3880±0.0038 | 0.00% ± 0.00% |
> |HighDim | 0.0000±0.0000 | -1.2724±1.2830 | 0.0000 ± 0.0000 | -197.2478±1.4391 | 0.00% ± 0.00% |
> | MISO | 0.0000±0.0000 | 0.2290±0.0103 | 0.0000 ± 0.0000 | 1.1495±0.0056 | 0.00% ± 0.00% |
>
> **Minor**
> **(1) Related works** We thank the reviewer's concern, and we will clarify the relevant descriptions and references.
>
> **(2) Compare with HomeoProj** We agree that this distinction should be stated more clearly.   In Section 2, we will add a concise clarification: (i) HomeoProj learns a parametric mapping, whereas HoP provides a closed-form, analytically defined homeomorphism. (ii) HomeoProj applies only to bounded feasible regions, while HoP naturally handles unbounded constraint sets.   (iii) HomeoProj still requires a post-projection to guarantee feasibility, whereas HoP is feasible by measure transformation without projection. The comparison table is provided as follows:
> ||HoP|HomeoProj|
> |-|-|-|
> |Homeomorphous Mapping|Invertible neural network (INN) and bisection (Implicit) |Polar measure transformation (Explicit)|
> |Feasibility Guarantee|Bisection Search with INN|Measure transformation|
> |Optimality|Slight Loss caused by distortion in INN|No loss|
> |Constraints Type|Bounded nonconvex(finite volume set)|Bounded, Semi-unbounded and non-convex sets where each ray has a single finite boundary intersection|
> **(3) Baseline of differentiable orthogonal projection**
> We appreciate the suggestion. In fact we think DC3 and HomeoProj are differentiable projection methods as well which share the similar design as differentiable orthogonal projection in PGD or FS-Net. However, these projection based methods rely on iterative searching. Although the orthogonal projection ensures the feasibility of result, the non-convexity of objective function still impacts result's optimality.
> **Questions**
> **(1) What constraints can HoP handle?**
> Hop can handle convex sets, star-convex sets, and non-convex sets with finite single ray–boundary intersections.
> **(2) Is DC3 truly post-correction?**
> We thank the reviewer for raising this concern. The answer depends on how post-correction is defined. In our paper, we treat any gradient-descent–based adjustment applied after the network’s raw output as post-correction, even if it is differentiable and used during end-to-end training. Our DC3 implementation follows the original method: the model predicts in the unconstrained space, and feasibility is enforced only afterward via gradient-based correction, which is also required at test time (given in the DC3 ablation). Thus, DC3 fundamentally relies on this correction. In contrast, HoP produces an angle and scale that map directly to feasible Cartesian coordinates and requires no gradient descent–based correction.

---

### Meta-Review · Area_Chair_kxBr · 2026-01-06

**Summary:**

This paper proposed Homeomorphic Polar Learning (HoP), a learning-to-optimize framework that embeds a homeomorphic mapping within neural networks to solve hard-constrained optimization problems by transforming latent polar representations into strictly feasible solutions. While it shows some merits, numerous reviewers' concerns are not fully addressed. Key unresolved issues include writing clarity; the computational complexity of determining the feasible boundary; unclear generalizability and robustness; and restricted applications to non-convex problems.

**Reviewer Concerns:**

### [7PFW]
All the concerns are addressed.

### [8LLu]

Weakness 1: Not fully addressed. While HoP theoretically supports star-convex sets, its efficacy relies heavily on identifying a valid center for the polar coordinate system. In the reported experiments, such as the $l_p$-norm problem, the center is trivially known (the origin). However, for general star-convex sets, identifying a center from which the boundary is star-convex is a computationally non-trivial task.

Weakness 2: Not fully addressed. While the authors clarify that their focus is on problem-specific L2O solvers rather than general-purpose backbones, this distinction does not exempt the method from the need for robustness and generalization. Even within a problem-specific context, it is unclear how well the proposed model generalizes to variations inherent to practical deployment, such as changes in problem scale (dimension), out-of-distribution (OOD) instances, or sensitivity to the interior-point initialization method ($y_0$).

Other concerns are addressed.

### [xWoM]

Weakness 2: Not fully addressed. The authors' claim that determining the boundary is a "1D root-finding problem regardless of dimension" is inaccurate. While the search space is indeed one-dimensional (along a ray), the evaluation cost at each step of that search is dimension-dependent. For instance, verifying feasibility against linear constraints ($Ax \leq b$) requires $\mathcal{O}(mn)$ operations. Furthermore, numerical stability issues, such as ill-conditioning, often scale with dimensionality, requiring additional iterations or precision in the root-finding process. Therefore, contrary to the rebuttal, the total computational cost remains inherently coupled to the problem dimension.

Weakness 3: Not fully addressed. The manuscript lacks clarity regarding the computational overhead of the proposed method. The critical procedures for computing the interior point ($y_0$) and the support radius ($\mathcal{R}$) are relegated to the appendix and are notably absent from Algorithm 1. This omission risks misleading readers into viewing these steps as trivial, whereas in practice, they constitute a significant portion of the computational burden.

Other concerns are addressed.

### [RKqY]

Weakness 2: Not fully addressed. While finding an interior point is sufficient and standard for convex sets, it is insufficient for non-convex cases (even relatively simple ones like star-convex sets). In such cases, one must identify a valid "center" for the polar mapping, which is a task that can incur substantial computational overhead. See previous comments.

Weakness 3: Not fully addressed. The authors' claim that determining the boundary is a "1D root-finding problem regardless of dimension" is inaccurate. See previous comments.

Question 2: Not addressed. The original question sought to understand the method's stability and sensitivity to different interior-point finding solvers. The rebuttal, however, focused on performance comparisons against SOTA methods using a fixed solver.

Others are addressed.

**Reviewer Scores:**

Reviewers 8LLu, xWoM, and RKqY retain their scores of 2. Although Reviewer 7PFW's concerns have been addressed, the AC anticipates that they will likely maintain their score of 6 given the context of the other three reviews.

---

### Decision · Program_Chairs · 2026-01-26

Reject